# Continuous MDP Homomorphisms and Homomorphic Policy Gradient

**Sahand Rezaei-Shoshtari**
McGill University and Mila

**Rosie Zhao**
McGill University and Mila

**Prakash Panangaden**
McGill University and Mila

**David Meger**
McGill University and Mila

**Doina Precup**
McGill University, Mila, and DeepMind

## Abstract

Abstraction has been widely studied as a way to improve the efficiency and generalization of reinforcement learning algorithms. In this paper, we study abstraction in the continuous-control setting. We extend the definition of MDP homomorphisms to encompass continuous actions in continuous state spaces. We derive a policy gradient theorem on the abstract MDP, which allows us to leverage approximate symmetries of the environment for policy optimization. Based on this theorem, we propose an actor-critic algorithm that is able to learn the policy and the MDP homomorphism map simultaneously, using the lax bisimulation metric. We demonstrate the effectiveness of our method on benchmark tasks in the DeepMind Control Suite. Our method's ability to utilize MDP homomorphisms for representation learning leads to improved performance when learning from pixel observations.

## 1   Introduction

For reinforcement learning from high-dimensional observations, such as images, learning a simpler problem by abstraction from the original problem can be critical [2, 49]. The coupling between states, actions and rewards complicates learning RL abstractions. MDP homomorphisms [64, 65, 67, 59] define a concept that allows one to exploit symmetries, yielding behavioral equivalence and preserving values, while giving the potential to arrive at a substantially smaller MDP. Recent works [81, 83, 11] have shown that using MDP homomorphisms is effective to guide learning in discrete problems. This paper is one of the first to consider MDP homomorphisms in the continuous-control setting and to develop actor-critic algorithms with a tightly integrated state-action abstraction. To that end, we identify and answer a series of key challenges:

**Can MDP homomorphisms be defined on continuous state and action spaces?** Our first contribution is

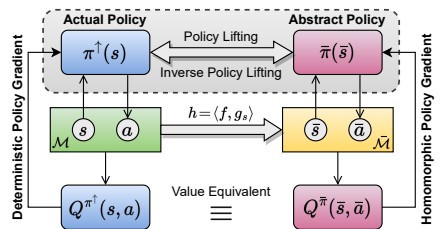

Figure 1: Schematics of our method. The actual MDP $\mathcal{M}$ is used to train $Q^{\pi^\uparrow}$ and update $\pi^\uparrow$ with DPG, while the abstract MDP $\overline{M}$ is used to train $Q^{\overline{\pi}}$ and update $\overline{\pi}$ with HPG. $\overline{\mathcal{M}}$ is the MDP homomorphic image of $\mathcal{M}$ obtained by learning the homomorphism map $h = (f, g_s)$. Policies $\pi^\uparrow$ and $\overline{\pi}$ can be derived from each other.

to define continuous MDP homomorphisms on continuous state and action spaces, which requires more intricate proofs that do not follow in any direct way from the finite case and requires tools from measure theory and differential geometry.

**Can MDP homomorphisms be tightly integrated into the policy gradient?** Our second contribution is the derivation of the *homomorphic policy gradient* (HPG) theorem to closely integrate the

abstract MDP into the policy gradient. Importantly, we rigorously prove that performing HPG on the abstract MDP is equivalent to performing the deterministic policy gradient (DPG) on the actual MDP. Therefore, HPG can act as an additional gradient estimator capable of utilizing approximate symmetries for improved sample efficiency. To derive these results, we prove that continuous MDP homomorphisms preserve value functions [33], which in turn enables their use for policy evaluation.

**Can MDP homomorphisms be learned simultaneously with the optimal policy in a practical deep reinforcement learning algorithm?** We propose a deep actor-critic algorithm, depicted in Figure 1, based on HPG, referred to as *Deep Homomorphic Policy Gradient* (DHPG), that unifies state and action abstractions. DHPG is able to simultaneously learn the policy and the homomorphism map using the lax bisimulation metric [78], a metric for measuring the equivalence of state-action pairs under an MDP homomorphism relation. We empirically show that state-action abstractions learned through MDP homomorphisms provide a natural inductive bias for representation learning.

Despite the existence of well-studied abstraction notions, learning state abstractions in a scalable fashion for continuous control remains a key challenge. In contrast to previous works on learning MDP homomorphisms [81, 83, 11], our algorithm is readily applicable to continuous actions, and compared to previous works guided by bisimulations [91, 29, 46], our algorithm leads to more robust solutions, as suggested by our empirical results. The bisimulation relation [56, 47, 12, 20, 30] and bisimulation metrics [21, 26, 25, 27] do not allow abstracting actions as they require exact matching of actions, whereas MDP homomorphisms and equivalently the lax bisimulation metric [78] remove this strong limitation, giving them greater modeling flexibility. Most importantly, the key difference between prior works [81, 83, 91, 29, 46] and our method is the homomorphic policy gradient theorem which allows for a tight integration of the abstraction notion into the policy gradient that theoretically motivates using the abstract MDP for policy optimization in the actual MDP. Our contributions are:

1. Defining continuous MDP homomorphisms on continuous state and action spaces, using tools from measure theory and differential geometry.
2. Proving that continuous MDP homomorphisms preserve value and optimal value functions.
3. Deriving the homomorphic policy gradient theorem.
4. Developing a deep actor-critic algorithm for learning the optimal policy simultaneously with the MDP homomorphism map in challenging continuous control problems.

DHPG improves upon strong baselines on pixel observations [88, 91] on DM Control, and our visualizations demonstrate the potential of MDP homomorphisms in learning structured representations that can preserve values and represent the minimal MDP image. To the best of our knowledge, this is the first homomorphic policy gradient derivation and the first work to define and scale up MDP homomorphisms to continuous visual control problems. Our code is publicly available at `https://github.com/sahandrez/homomorphic_policy_gradient`.

# 2  Background

## 2.1  Markov Decision Processes

We consider the standard MDP that is defined by a 5-tuple $(\mathcal{S}, \mathcal{A}, \tau_a, R, \gamma)$, with *state space* $\mathcal{S}$, *action space* $\mathcal{A}$, *transition dynamics* $\tau_a : \mathcal{S} \times \mathcal{A} \to \mathrm{Dist}(\mathcal{S})$, *reward function* $R : \mathcal{S} \times \mathcal{A} \to \mathbb{R}$, and *discount factor* $\gamma \in (0, 1]$. The goal is to find a policy $\pi : \mathcal{S} \to \mathrm{Dist}(\mathcal{A})$ that maximizes the expected sum of discounted rewards, the *expected return*, defined as $\mathbb{E}_\pi[R_t] = \mathbb{E}_\pi[\sum_{k=0}^{T} \gamma^k r_{t+k+1}]$. *Value function* $V^\pi(s)$ denotes the expected return from $s$ under policy $\pi$, and *action-value function* $Q^\pi(s, a)$ denotes the expected return from $s$ after taking action $a$ under $\pi$. Value functions are fixed points of the Bellman equation [8] and can be computed iteratively through a process referred to as *policy evaluation* [75]. Similarly, optimal value functions $V^*(s)$ and $Q^*(s, a)$ are fixed points of the Bellman optimality equation [8].

## 2.2  MDP Homomorphisms

MDP homomorphisms are formally defined for finite MDPs by Ravindran and Barto [65] as:

**Definition 1** (MDP Homomorphism). An *MDP homomorphism* $h = (f, g_s) : \mathcal{M} \to \overline{\mathcal{M}}$ is a surjective map from a finite MDP $\mathcal{M} = (\mathcal{S}, \mathcal{A}, R, \tau_a, \gamma)$ onto an abstract finite MDP $\overline{\mathcal{M}} = (\overline{\mathcal{S}}, \overline{\mathcal{A}}, \overline{R}, \overline{\tau}_{\overline{a}}, \gamma)$ where

$f \colon \mathcal{S} \to \overline{\mathcal{S}}$ and $g_s \colon \mathcal{A} \to \overline{\mathcal{A}}$ are surjective maps onto the abstract state and action spaces:

$$\text{Invariance of reward: } \overline{R}(f(s), g_s(a)) = R(s, a) \quad \forall s \in \mathcal{S}, a \in \mathcal{A} \tag{1}$$

$$\text{Equivariance of transitions: } \overline{\tau}_{g_s(a)}(f(s')|f(s)) = \sum_{s'' \in [s']_{B_h|\mathcal{S}}} \tau_a(s''|s) \quad \forall s \in \mathcal{S}, a \in \mathcal{A} \tag{2}$$

where $B_h$ is the partition of $\mathcal{S}$ induced by the equivalence relation of homomorphism $h$, $B_h|\mathcal{S}$ is the projection of $B_h$ onto $\mathcal{S}$, and $[s']_{B_h|\mathcal{S}}$ denotes the block of $B_h|\mathcal{S}$ to which $s'$ belongs. Thus, when applying action $a$ in state $s$, the right-hand side is the probability that the resulting state is in $[s']_{B_h|\mathcal{S}}$. The abstract MDP $\overline{\mathcal{M}}$ is in fact the quotient MDP $\mathcal{M}/B_h$ based on the homomorphism map $h \colon \mathcal{M} \to \mathcal{M}/B_h$. As MDP homomorphisms are sensitive with respect to changes in rewards or transitions, *approximate* MDP homomorphisms [67] allow equations (1-2) to hold approximately. The significance of MDP homomorphisms is the *optimal value equivalence* between $\mathcal{M}$ and $\overline{\mathcal{M}}$ [65]:

$$V^*(s) = \overline{V}^*(f(s)) \quad \forall s \in \mathcal{S}, \qquad Q^*(s, a) = \overline{Q}^*(f(s), g_s(a)) \quad \forall s \in \mathcal{S}, a \in \mathcal{A} \tag{3}$$

which in turn allows for learning the optimal policy $\overline{\pi}^*$ in the abstract MDP and consequently *lifting* it to obtain the optimal policy in the actual MDP, using:

$$\pi^\uparrow(a|s) = \frac{\overline{\pi}(\overline{a}|f(s))}{|\{a \in g_s^{-1}(\overline{a})\}|}, \qquad \forall s \in \mathcal{S}, a \in g_s^{-1}(\overline{a})$$

where $g_s^{-1}(\overline{a})$ denotes the set of actions that have the same image $\overline{a}$ under $g_s$. Equivalently, the two policies must satisfy $\sum_{a \in g_s^{-1}(\overline{a})} \pi^\uparrow(a|s) = \overline{\pi}(\overline{a}|f(s))$ for all $s \in \mathcal{S}$ and $\overline{a} \in \overline{\mathcal{A}}$.

## 2.3 Bisimulation and Lax Bisimulation Metrics

*Bisimulation* for finite MDPs [20, 30] defines an equivalence relation on $\mathcal{S}$ where two states $s_i$ and $s_j$ are equivalent or *bisimilar* if $R(s_i, a) = R(s_j, a)$ and $\tau_a(C|s_i) = \tau_a(C|s_j)$ for all $a \in \mathcal{A}$ and every equivalence class $C$ defined by the equivalence relation. The rigidity of bisimulation limits its applications. *Bisimulation metrics* [26, 25, 27] measure the equivalence as an approximation:

$$d_{\text{bisim}}(s_i, s_j) = \max_{a \in \mathcal{A}} c_r |R(s_i, a) - R(s_j, a)| + c_t K(\tau_a(\cdot|s_i), \tau_a(\cdot|s_j)), \tag{4}$$

where the first term measures reward similarity and $K$ is the Kantorovich (Wasserstein) metric measuring the distance between the transition probabilities. However, bisimulation metrics can still be brittle as they require the behaviour to match for all actions. This may be problematic particularly in the case of continuous actions in which small changes to actions may not drastically change the outcome. Additionally, bisimulation metrics are not able to represent environment symmetries. Instead, *lax bisimulation* [78] waives the requirement on action matching in favor of extending the state equivalence relation to state-action equivalence. Taylor et al. [78] show that lax bisimulation is precisely the same relation as the MDP homomorphism and define the *lax bisimulation metric* as:

$$d_{\text{lax}}((s_i, a_i), (s_j, a_j)) = c_r |R(s_i, a_i) - R(s_j, a_j)| + c_t K(\tau_{a_i}(\cdot|s_i), \tau_{a_j}(\cdot|s_j)). \tag{5}$$

Furthermore, Taylor et al. [78] show that minimizing the lax bisimulation metric corresponds to finding approximate MDP homomorphisms and bound the value error.

## 3 Value Equivalence Property

To motivate the use of MDP homomorphisms for policy evaluation and consequently policy optimization, we first prove their *value equivalence property* in the finite case as the generalization of the prior result of the *optimal* value equivalence [65], stated in Equation (3). The proof is in Appendix C.1.

**Theorem 1** (Value Equivalence)**.** *Let $\overline{\mathcal{M}}$ be the image of an MDP homomorphism $h$ from a finite $\mathcal{M}$. Then any two corresponding policies $\pi^\uparrow = \text{lift}(\overline{\pi})$ have equivalent values:*

$$V^{\pi^\uparrow}(s) = V^{\overline{\pi}}(f(s)) \quad \forall s \in \mathcal{S}, \qquad Q^{\pi^\uparrow}(s, a) = Q^{\overline{\pi}}(f(s), g_s(a)) \quad \forall s \in \mathcal{S}, a \in \mathcal{A}$$

# 4 Continuous MDP Homomorphisms

To concretely lay out the foundations of using MDP homomorphisms for continuous control, we extend their definition to continuous state and action spaces, and derive results analogous to the finite case. First, we define continuous MDPs and state our underlying assumptions. Importantly, the correct definitions of continuous MDPs and continuous MDP homomorphisms require care regarding measurability and differentiability of spaces, and our formulation is chosen to fit the HPG derivation; see Appendix A.2 for an overview of the tools we used from measure theory and differential geometry.

**Definition 2** (Continuous MDP). A *continuous Markov decision process (MDP)* is a 6-tuple:

$$\mathcal{M} = (\mathcal{S}, \Sigma, \mathcal{A}, \forall a \in \mathcal{A} \; \tau_a : \mathcal{S} \times \Sigma \to [0,1], R : \mathcal{S} \times \mathcal{A} \to \mathbb{R}, \gamma)$$

where $\mathcal{S}$, the state space is assumed to be a Polish space, $\Sigma$ is a $\sigma$-algebra on $\mathcal{S}$[1], $\mathcal{A}$, the space of *actions*, is a locally compact metric space, usually taken to be a subset of $\mathbb{R}^n$, $\tau_a$ is the transition probability kernel for each possible action $a$, for each fixed $s$, $\tau_a(\cdot|s)$ is a probability distribution on $\Sigma$ while $R$ is the reward function, and $\gamma$ is the discount factor. Furthermore, for all $s \in \mathcal{S}$ and $B \in \Sigma$ the map $a \mapsto \tau_a(B|s)$ is smooth. The last assumption is required for differentiability with respect to actions $a$, which is needed in Section 5 for deriving the HPG theorem.

Given the continuous MDPs described above, we define continuous MDP homomorphisms. The equivariance condition on the transition dynamics, Equation (2), can no longer be expressed in terms of a discrete sum over partitions, and instead we use the $\sigma$-algebra structure on the different state spaces.

**Definition 3** (Continuous MDP Homomorphism). A *continuous MDP homomorphism* is a map $h = (f, g_s) : \mathcal{M} \to \overline{\mathcal{M}}$ where $f : \mathcal{S} \to \overline{\mathcal{S}}$ and for every $s$ in $\mathcal{S}$, $g_s : \mathcal{A} \to \overline{\mathcal{A}}$ are measurable, surjective maps such that the following hold:

$$\text{Invariance of reward: } \overline{R}(f(s), g_s(a)) = R(s, a) \qquad \forall s \in \mathcal{S}, a \in \mathcal{A} \tag{6}$$

$$\text{Equivariance of transitions: } \overline{\tau}_{g_s(a)}(\overline{B}|f(s)) = \tau_a(f^{-1}(\overline{B})|s) \qquad \forall \; s \in \mathcal{S}, a \in \mathcal{A}, \overline{B} \in \overline{\Sigma} \tag{7}$$

Note that if $g_s$ is the identity map, the second condition reduces to $\overline{\tau}_a(\overline{B}|f(s)) = \tau_a(f^{-1}(\overline{B})|s)$ which is simply the condition for preservation of transition probabilities as used in bisimulation [20].

## 4.1 Optimal Value Equivalence

Assuming the conditions given in Definition 3, we prove that optimal value functions are preserved by the continuous MDP homomorphism as in the finite case.

**Theorem 2** (Optimal Value Equivalence). *Let $\overline{\mathcal{M}} = (\overline{\mathcal{S}}, \overline{\Sigma}, \overline{\mathcal{A}}, \overline{\tau}_{\overline{a}}, \overline{R})$ be the image of a continuous MDP homomorphism $h = (f, g_s)$ from $\mathcal{M} = (\mathcal{S}, \Sigma, \mathcal{A}, \tau_a, R)$. Then:*

$$V^*(s) = \overline{V}^*(f(s)) \quad \forall s \in \mathcal{S}, \qquad Q^*(s,a) = \overline{Q}^*(f(s), g_s(a)) \quad \forall (s,a) \in \mathcal{S} \times \mathcal{A} \tag{8}$$

The proof, given in Appendix C.2, uses the change of variable formula of the pushforward measure of $\tau_a(\cdot|s)$ with respect to $f$ to change the integration space from $\mathcal{S}$ to $\overline{\mathcal{S}}$.

## 4.2 Value Equivalence for Lifting Deterministic Policies

As in the finite case, we also require a lifting process to define $\pi^\uparrow = \mathit{lift}(\overline{\pi})$ given a policy $\overline{\pi}$ on the abstract MDP. In general, the lifted policy needs to satisfy the relation $\pi^\uparrow(g_s^{-1}(\beta)|s) = \overline{\pi}(\beta|f(s))$ for every Borel set $\beta \subseteq \overline{\mathcal{A}}$ and $s \in \mathcal{S}$. While our initial progress shows that lifted stochastic policies exist based on the disintegration theorem, the full proof and design of a computationally tractable algorithm for this process is left for future work. Therefore, here and in the subsequent sections we assume the policy is deterministic in which case the lifted policy can be simply obtained by choosing one representative for the preimage $g_s^{-1}(\overline{\pi}(f(s)))$. If we select $g_s$ to be a bijection, the lifted policy can be uniquely defined as $\pi^\uparrow(s) = g_s^{-1}(\overline{\pi}(f(s)))$. The assumption on deterministic policies is not limiting, as in general the optimal policy of a given MDP is deterministic [9]. With this lifting definition, we state and prove the following value equivalence result:

---

[1] Usually the Borel algebra.

**Theorem 3** (Value Equivalence for Deterministic Policies). *Let $\overline{\mathcal{M}}$ be the image of a continuous MDP homomorphism $h = (f, g_s)$ from $\mathcal{M}$, then any two deterministic policies $\pi^{\uparrow} : \mathcal{S} \to \mathcal{A}$ and $\overline{\pi} : \overline{\mathcal{S}} \to \overline{\mathcal{A}}$ where $\pi^{\uparrow} = \text{lift}(\overline{\pi})$ have equivalent value functions on their domain:*

$$V^{\pi^{\uparrow}}(s) = V^{\overline{\pi}}(f(s)) \quad \forall s \in \mathcal{S}, \qquad Q^{\pi^{\uparrow}}(s,a) = Q^{\overline{\pi}}(f(s), g_s(a)) \quad \forall (s,a) \in \mathcal{S} \times \mathcal{A}$$

The proof, given in Appendix C.3, uses the change of variable formula of the pushforward measure of $\tau_a(\cdot|s)$ with respect to $f$ to change the integration space from $\mathcal{S}$ to $\overline{\mathcal{S}}$ and assumes $g_s$ to be bijective.

## 5 Homomorphic Policy Gradient

The next goal of this work is to derive a policy gradient estimator using samples obtained from the abstract MDP. Intuitively, this allows for direct incorporation of state-action abstraction as an inductive bias for policy optimization, thereby reducing the variance of actor updates and improving sample efficiency. Equipped with continuous MDP homomorphisms from Definition 3 and their value equivalence property, we now derive the *homomorphic policy gradient* (HPG) theorem.

In this section, we assume having access to an MDP homomorphism map $h = (f, g_s)$, parameterized by differentiable functions. The problem of learning such mapping from samples is addressed in Section 6. Additionally, we assume the MDP and the homomorphism map adhere to the conditions of Definition 2 and Appendix B. Similarly to prior works on policy gradients [76, 70], we define the performance measure as $J(\theta) = \mathbb{E}_{\pi}[V^{\pi}(s)]$ where the expectation is over the uncertainty in transitions, rewards, and initial states. Finally, as detailed in Section 4.2, our results are derived for deterministic policies and a bijective $g_s$. Notably, this choice allows us to parameterize one of the policies and to uniquely derive the other policy. In practice, we parameterize the actual policy as $\pi_{\theta}^{\uparrow}$ and obtain the abstract policy as $\overline{\pi}_{\theta} = g_s(\pi_{\theta}^{\uparrow}(s))$. First, we show the *equivalence of policy gradients*:

**Theorem 4** (Equivalence of Deterministic Policy Gradients). *Let $\overline{\mathcal{M}}$ be the image of a continuous MDP homomorphism $h$ from $\mathcal{M}$, and let $\pi_{\theta}^{\uparrow} : \mathcal{S} \to \mathcal{A}$ be the lifted deterministic policy corresponding to the abstract deterministic policy $\overline{\pi}_{\theta} : \overline{\mathcal{S}} \to \overline{\mathcal{A}}$. Then for any $(s,a) \in \mathcal{S} \times \mathcal{A}$ we have:*

$$\nabla_a Q^{\pi_{\theta}^{\uparrow}}(s,a)\Big|_{a=\pi_{\theta}^{\uparrow}(s)} \nabla_{\theta} \pi_{\theta}^{\uparrow}(s) = \nabla_{\overline{a}} Q^{\overline{\pi}_{\theta}}(\overline{s}, \overline{a})\Big|_{\overline{a}=\overline{\pi}_{\theta}(\overline{s})} \nabla_{\theta} \overline{\pi}_{\theta}(\overline{s}).$$

The proof is given in Appendix C.4 and uses the chain rule and the inverse function theorem on manifolds, which in turn raises the need for $g_s$ to be a bijection and local diffeomorphism. Theorem 4 highlights that the gradient of the abstract MDP is equivalent to that of the original, despite the underlying spaces being abstracted. This implies that performing HPG on the abstract MDP is equivalent to performing DPG on the actual MDP, allowing us to use them synergistically to update the same parameters $\theta$, as shown in Figure 1.

While one can naively use Theorem 4 to substitute gradients of the standard DPG, theoretically this does not produce any useful result as the expectation remains estimated with respect to the stationary state distribution of the actual MDP $\mathcal{M}$ under $\pi_{\theta}^{\uparrow}(s)$. However, using properties of continuous MDP homomorphisms, we can change the integration space from $\mathcal{S}$ to $\overline{\mathcal{S}}$, and consequently estimate the policy gradient with respect to the stationary distribution of the abstract MDP $\overline{\mathcal{M}}$ under $\overline{\pi}_{\theta}(\overline{s})$:

**Theorem 5** (Homomorphic Policy Gradient Theorem). *Let $\overline{\mathcal{M}}$ be the image of a continuous MDP homomorphism $h$ from $\mathcal{M}$, and let $\overline{\pi}_{\theta} : \overline{\mathcal{S}} \to \overline{\mathcal{A}}$ be a deterministic abstract policy defined on $\overline{\mathcal{M}}$. Then the gradient of the performance measure $J(\theta)$, defined on the actual MDP $\mathcal{M}$, w.r.t. $\theta$ is:*

$$\nabla_{\theta} J(\theta) = \int_{\overline{s} \in \overline{\mathcal{S}}} \rho^{\overline{\pi}_{\theta}}(\overline{s}) \nabla_{\overline{a}} Q^{\overline{\pi}_{\theta}}(\overline{s}, \overline{a})\Big|_{\overline{a}=\overline{\pi}_{\theta}(\overline{s})} \nabla_{\theta} \overline{\pi}_{\theta}(\overline{s}) d\overline{s}.$$

*where $\rho^{\overline{\pi}_{\theta}}(\overline{s})$ is the discounted state distribution of $\overline{\mathcal{M}}$ following the deterministic policy $\overline{\pi}_{\theta}(\overline{s})$.*

The proof is given in Appendix C.5 and applies the result of Theorem 4 and the change of variables formula of the pushforward measure on the state space. The significance of Theorem 5, which forms the basis of our proposed homomorphic actor-critic algorithm, is twofold. First, we can get another estimate for the policy gradient based on the approximate MDP homomorphic image in addition to DPG. Although the two policy gradient estimates are not statistically independent from one another

as they are tied through the homomorphism map, HPG will potentially have less variance at the expense of some bias due to the approximation of the MDP homomorphism.

Second, since the minimal image of an MDP is the MDP homomorphic image [65], the abstract critic $Q^{\overline{\pi}_\theta}$ is trained on a simplified problem. In other words, each abstract state-action pair $(\overline{s}, \overline{a})$ used to train $Q^{\overline{\pi}_\theta}$ represents all $(s, a)$ pairs that are equivalent under the MDP homomorphism relation, thus improving sample efficiency. However, the amount of complexity reduction is dependent on the approximate symmetries of the environment, as also supported by our empirical results.

## 6  Homomorphic Actor-Critic Algorithms

We propose a deep actor-critic algorithm based on HPG by adapting DDPG [51]. We refer to our method as *Deep Homomorphic Policy Gradient* (DHPG). Although HPG is applicable to any other deterministic actor-critic, we chose DDPG as its simplicity compared to modern choices [7, 43] allows for a better study on the impact of MDP homomorphisms. Notably, a strong advantage of DHPG is that it is readily applicable to pixel observations without the need for extra mechanisms such as image reconstruction [29, 34, 90, 36], as the notion of MDP homomorphism provides a natural inductive bias for learning representations that preserve values and optimal values.

Since DHPG is learning the MDP homomorphism map $h$ online and concurrently with the policy, using the actual MDP for training the critic, specifically at the early stages of training, is helpful. Therefore, we utilize a separate critic for each MDP $\mathcal{M}$ and $\overline{\mathcal{M}}$. Ultimately, critics are used to update a single set of parameters, as shown in Figure 1; see Appendix D.5 for the ablation study on this.

Thus, the components of DHPG are: actual critic $Q_\psi(s, a)$, abstract critic $\overline{Q}_{\overline{\psi}}(\overline{s}, \overline{a})$, deterministic actor $a = \pi_\theta(s)$, homomorphism map $h_{\phi,\eta} = (f_\phi(s), g_\eta(s, a))$, reward predictor $\overline{R}_\rho(\overline{s})$, and probabilistic transition model $\overline{\tau}_\nu(\overline{s}'|\overline{s}, \overline{a})$ which outputs a Gaussian distribution. We use target networks and a vanilla replay buffer [57, 51]. As discussed in Section 5, an abstract actor is obtained as $\overline{a} = g_s(\pi_\theta(s))$. In case of pixel observations, a single image encoder $E_\mu$ is shared among all components.

**Training the Policy and Critic.**  Actual and abstract critics are trained using $n$-step TD error for a faster reward propagation [7]. The loss function for each critic is therefore defined as the expectation of the $n$-step Bellman error estimated over transitions samples from the replay buffer $\mathcal{B}$:

$$\mathcal{L}_{\text{actual critic}}(\psi) = \mathbb{E}_{(s,a,s',r)\sim\mathcal{B}}\Big[\big(R_t^{(n)} + \gamma^n Q_{\psi'}(s_{t+n}, a_{t+n}) - Q_\psi(s_t, a_t)\big)^2\Big] \qquad (9)$$

$$\mathcal{L}_{\text{abstract critic}}(\overline{\psi}, \phi, \eta) = \mathbb{E}_{(s,a,s',r)\sim\mathcal{B}}\Big[\big(R_t^{(n)} + \gamma^n \overline{Q}_{\overline{\psi}'}(\overline{s}_{t+n}, \overline{a}_{t+n}) - \overline{Q}_{\overline{\psi}}(\overline{s}_t, \overline{a}_t)\big)^2\Big], \qquad (10)$$

where $\overline{s}_t = f_\phi(s_t)$ and $\overline{a}_t = g_\eta(s_t, a_t)$ are computed using the learned MDP homomorphism, $\psi'$ and $\overline{\psi}'$ denote parameters of target networks, and $R_t^{(n)} = \sum_{i=0}^{n-1} \gamma^i r_{t+i}$ is the $n$-step return. Consequently, we train the policy using DPG [70] and HPG from Theorem 5 by backpropagating the following loss:

$$\mathcal{L}_{\text{actor}}(\theta) \approx -\mathbb{E}_{s\sim\mathcal{B}}\Big[Q_\psi\big(s, \pi_\theta(s)\big) + \overline{Q}_{\overline{\psi}}\big(f_\phi(s), g_\eta(s, \pi_\theta(s))\big)\Big]. \qquad (11)$$

Here, the two gradients are added together and a single policy update is conducted; see Appendix D.5 for the ablation study on other combinations of HPG and DPG. Finally, we utilize target policy smoothing and delayed actor updates [28]. The pseudo-code of DHPG is presented in Appendix E.1.

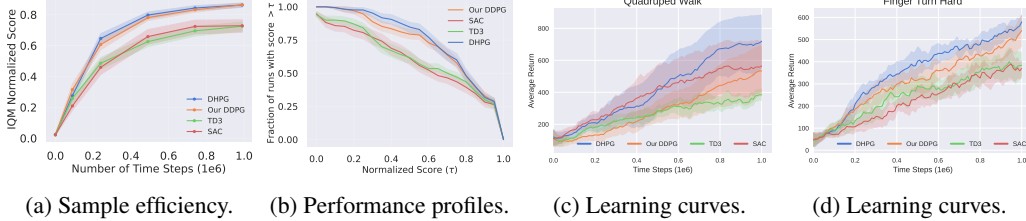

(a) Sample efficiency.  (b) Performance profiles.  (c) Learning curves.  (d) Learning curves.

Figure 2: Results of DM Control tasks with **state observations** obtained on 10 seeds. RLiable metrics are aggregated over 17 tasks. **(a)** RLiable IQM scores as a function of number of steps for comparing sample efficiency, **(b)** RLiable performance profiles at 500k steps, **(c)-(d)** examples of learning curves. Full results are in Appendix D.1. Shaded regions represent 95% confidence intervals.

**Learning Continuous MDP Homomorphisms.** We now address the problem of learning continuous MDP homomorphisms. While few methods have been proposed for learning finite MDP homomorphisms [81, 83], these are not readily extendable to continuous actions. In this work, we use the lax bisimulation metric [78], Equation (5), to propose a loss function that encodes lax bisimilar states closer together in the abstract space. The lax bisimulation metric is applicable to continuous actions and as a (pseudo-)metric, it can naturally represent approximate MDP homomorphisms.

Following the same intuition as prior works on bisimulations [91], we define our proposed lax bisimulation loss over pairs of transition tuples sampled from the replay buffer. We permute samples to compute their pairwise distance in the abstract space and their pairwise lax bisimilarity distance. Consequently, we minimize the distance between these two terms:

$$\mathcal{L}_{\text{lax}}(\phi, \eta) = \mathbb{E}_{\mathcal{B}}\big[\|f_\phi(s_i) - f_\phi(s_j)\|_1 - \|r_i - r_j\|_1 - \alpha W_2\big(\overline{\tau}_\nu(\cdot|f_\phi(s_i), g_\eta(s_i, a_i)), \overline{\tau}_\nu(\cdot|f_\phi(s_j), g_\eta(s_j, a_j))\big)\big]$$
(12)

Similarly to Zhang et al. [91], we replaced the Kantorovich ($W_1$) metric in Equation (5) with the $W_2$ metric as there is an explicit formula for it for Gaussian distributions. Finally, we apply the conditions of a continuous MDP homomorphism map from Definition 3 via the loss function of:

$$\mathcal{L}_{\text{h}}(\phi, \eta, \nu, \rho) = \mathbb{E}_{(s_i, a_i, s_i', r_i) \sim \mathcal{B}}\big[\big(f_\phi(s_i') - \overline{s}_i'\big)^2 + \big(r_i - \overline{R}_\rho(f_\phi(s_i))\big)^2\big],$$
(13)

where $\overline{s}_i' \sim \overline{\tau}_\nu(\cdot|f_\phi(s_i), g_\eta(s_i, a_i))$. The final loss function is obtained as $\mathcal{L}_{\text{lax}}(\phi, \eta) + \mathcal{L}_{\text{h}}(\phi, \eta, \rho, \nu)$.

# 7 Experiments

In our experiments, we aim to answer the following key questions:

1. Does the homomorphic policy gradient improve policy optimization?
2. What are the qualitative properties of the learned representations and the abstract MDP?
3. Can DHPG learn and recover the minimal MDP image from raw pixel observations?

We evaluate DHPG on continuous control tasks from DM Control on state and pixel observations. Importantly, to reliably evaluate our algorithm against the baselines and to correctly capture the distribution of results, we follow the best practices proposed by Agarwal et al. [5] and report the interquartile mean (IQM) and performance profiles aggregated on all tasks over 10 random seeds. While our baseline results are obtained using the official code, when possible [2], some of the results may differ from the originally reported ones due to the difference in the seed numbers and our goal to present a faithful representation of the true performance distribution [5].

## 7.1 State Observations

We compare DHPG on state observations against three commonly-used off-policy model-free algorithms: DDPG [51], TD3 [28], and SAC [35]. All methods use $n$-step returns, and share the same hyperparameters presented in Appendix E.2. For a fair comparison with DDPG and a better study of the impact of HPG, we have improved our DDPG by adding delayed policy updates and target policy [28]. Thus, the only difference between our DDPG and TD3 is the clipped double Q-learning present in TD3, which appears to be hurting the performance in some tasks of DMC as also observed in [60, 50].

**DHPG outperforms or matches other algorithms on state observations and has a better sample efficiency.** Results are presented in Figure 2, and *full results are in Appendix D.1.* Expectedly, performance gains are larger on tasks with symmetries, as DHPG is able to learn a compressed abstract MDP.

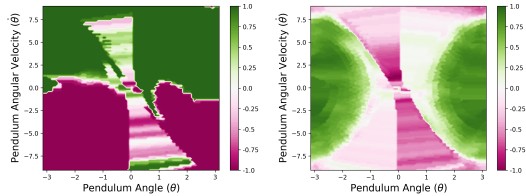

(a) Actual optimal policy.  (b) Abstract optimal policy.

Figure 3: Contours of actual and abstract optimal actions over the state space of the pendulum-swingup task. Colors represent action values, and states are $s = (\theta, \dot{\theta})$. **(a)** Actual optimal policy; contours of optimal actions $a^* = \pi^{\uparrow^*}(s)$. **(b)** Abstract optimal policy; contours of abstract optimal actions $\overline{a}^* = g_s(a^*) = \overline{\pi}^*(\overline{s})$. The relation $g_{s_1}(a_1) = g_{s_2}(a_2)$ holds for equivalent state-action pairs, and the abstract optimal policy is symmetric.

---

[2] We use the official implementations of DrQv2, DBC, and SAC-AE, while we re-implement DeepMDP due to the unavailability of the official code. See Appendix E.3 for full details.

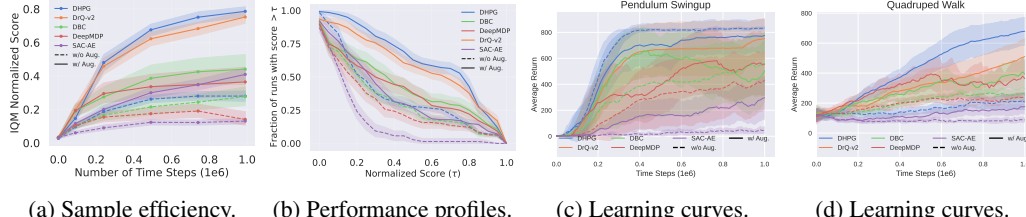

(a) Sample efficiency.  (b) Performance profiles.  (c) Learning curves.  (d) Learning curves.

Figure 4: Results of DM Control tasks with **pixel observations** obtained on 10 seeds. RLiable metrics are aggregated over 14 tasks. **(a)** RLiable IQM scores as a function of number of steps for comparing sample efficiency, **(b)** RLiable performance profiles at 500k steps, **(c)**-**(d)** examples of learning curves. Full results are in Appendix D.2. Shaded regions represent $95\%$ confidence intervals.

**The learned mapping** $h=(f, g_s)$ **demonstrates properties of an MDP homomorphism.** We use the pendulum swingup task to visualize its learned MDP homomorphism, as its symmetries are perfectly intelligible. Two state-action pairs $(s_1 = (\theta_1, \dot{\theta}_1), a_1)$ and $(s_2 = (\theta_2, \dot{\theta}_2), a_2)$ are equivalent if $a_1 = -a_2$, $\theta_1 = -\theta_2$, and $\dot{\theta}_1 = -\dot{\theta}_2$. Therefore, the learned action representations are expected to reflect this by setting $g_{s_1}(a_1) = g_{s_2}(a_2)$. Figure 3a shows contours of optimal actions over $\mathcal{S}$, while Figure 3b shows action representations $\bar{a} = g_s(a)$ of optimal actions over $\mathcal{S}$. Clearly, abstract actions adhere to the aforementioned relation for equivalent state-action pairs, indicating $g_s(a)$ is in fact representing the action encoder of an MDP homomorphism mapping.

## 7.2 Pixel Observations

We compare the effectiveness of DHPG on pixel observations against DBC [91], DeepMDP [29], SAC-AE [90], and state-of-the-art performing DrQ-v2 [88]. All methods use $n$-step returns, share the same hyperparameters in Appendix E.2 and all hyperparameters are adapted from DrQ-v2 *without any further tuning*. Importantly, for a fair comparison with DrQ-v2 which uses image augmentation, we present two variations of DHPG and other baselines, *with and without image augmentation*.

**DHPG outperforms or matches other algorithms on pixel observations, demonstrating its effectiveness in representation learning.** Results are presented in Figure 4 and *full results are in Appendix D.2*. Interestingly, DHPG without image augmentation outperforms DrQ-v2 on domains with easily learnable MDP homomorphism maps, such as cartpole and pendulum, showing its power of representation learning.

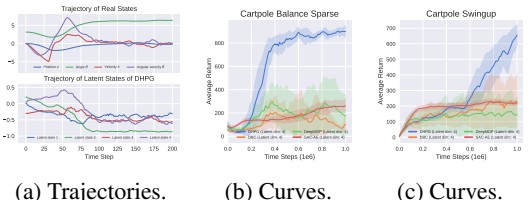

(a) Trajectories.  (b) Curves.  (c) Curves.

Figure 5: Effectiveness of DHPG in recovering the minimal MDP from **pixels**. All methods are limited to a 4-dimensional latent space which is equal to the dimensions of the real state space of cartpole. **(a)** Trajectories of real states obtained from Mujoco and trajectories of latent states of DHPG. **(b, c)** Learning curves averaged on 10 seeds.

**DHPG can learn and recover a low-dimensional MDP image.** A key strength of MDP homomorphisms is their ability to represent the minimal MDP image [65], which is particularly important when learning from pixel observations. To demonstrate this ability, we have limited the latent space dimensions to the dimension of the real system and compared DHPG (without image augmentation) with baselines in Figure 5. While other methods are not able to learn the tasks, DHPG can successfully learn the policy and the minimal low-dimensional latent space. Surprisingly, trajectories of the latent states resemble that of the real states as shown in Figure 5a.

**The abstract MDP demonstrates properties of an MDP homomorphic image.** To qualitatively demonstrate the significance of learning joint state-action representations, Figure 6 shows visualizations of latent states for quadruped-walk, a task with symmetries around movements of its four legs. Interestingly, while the latent space of DHPG (Figure 6a) shows distinct states for each leg, abstract state encoder $f_\phi$ has mapped corresponding legs (e.g., left forward leg and right backward leg) to the same abstract latent state (Figure 6b) as they are some homomorphic image of one another. Clearly, DBC and DrQ-v2 are not able to achieve this.

**The learned representations and the MDP homomorphism map transfer to new tasks within the same domain.** Importantly, one consideration with representation learning methods relying on rewards is the transferability of the learned representations to a new reward setting within the same domain. To ensure that our method does not hinder such transfer, we have carried out experiments in which the actor, critics, and the learned MDP homomorphism map are transferred to another task from the same domain. Results, given in Appendix D.3 show that our method has not compromised transfer abilities.

**Accounting for the larger network capacity of DHPG compared to the baselines.** Since our DHPG algorithm contains additional networks, such as the parameterized MDP homomorphism map and the abstract critic, it may have a higher network capacity compared to the baselines. To control for the effect of the network capacity and for a fair evaluation, we compare DHPG with higher-capacity variants of DBC and DrQ-v2 that have a larger critic networks, selected such that the total number of parameters are considerably more than that of DHPG. Results are presented in Figure 7, while *full results and a detailed description of the total number of parameters are in Appendix D.7*. As suggested by the results, DHPG outperforms or matches the performance of the higher-capacity baselines, demonstrating the improved performance is rather due to the use of the abstract MDP homomorphic image for representation learning and performing HPG updates.

**Additional Experiments.** We study the value equivalence property as a measure for the quality of the learned MDP homomorphisms in Appendix D.4, and we present ablation studies on DHPG variants, and the impact of $n$-step return on our method in Appendices D.5 and D.6, respectively.

## 8   Related Work

**State Abstraction.** Bisimulation [56, 47] is a notion of behavioral equivalence between systems. It was extended to continuous state spaces by Blute et al. [12, 20] and extended to MDPs by Givan et al. [30]. Bisimulation metrics [21, 26, 25, 27] define a pseudometric to quantify the degree of behavioural similarity. Recently, Zhang et al. [91] defined a loss function for learning representations via bisimilarity of latent states, and Kemertas et al. [46] have further improved its robustness. Castro [14] has proposed a method to approximate the bisimulation metric for deterministic MDPs with continuous states but discrete actions. van der Pol et al. [81] have defined a contrastive loss based on MDP homomorphisms for learning an abstract MDP for planning, however, their method is only applicable to finite MDPs. Another approach is to directly embed the MDP homomorphic relation in the network architecture [83, 82]. Other recently proposed methods seek to learn representations that preserve values [33, 32] or policies [4], or via a sampling-based similarity metric [15]. Finally, state abstractions can in principle help improve transferring of policies [1, 16, 72, 73, 63], or learning temporally extended actions [17, 87, 86, 77].

**Action Abstraction.** Action representations are often studied in the context of large discrete action spaces [68] as a form of a look-up embedding that is known *a-priori* [22], factored representations [69], or policy decomposition [18]. Action representations can also be learned from expert demonstrations [79]. More related to our work is dynamics-aware embeddings [85] where a combined state-action

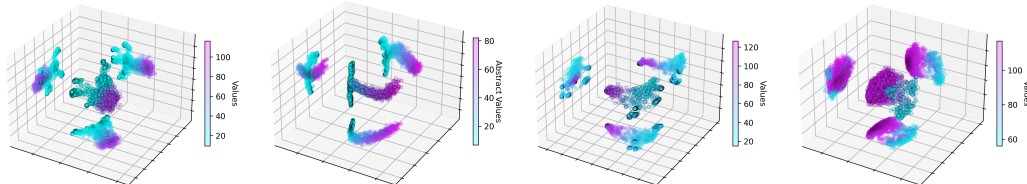

(a) Latent states, DHPG.    (b) Abstract states, DHPG.    (c) Latent states, DBC.    (d) Latent States, DrQ-v2.

Figure 6: PCA projection of learned representations for quadruped-walk with **pixel observations**. **(a)** Latent states $s = E_\mu(o)$, **(b)** abstract latent states $\overline{s} = f_\phi(E_\mu(o))$ for DHPG, **(c)** latent states $s = E_\mu(o)$ for DBC, and **(d)** DrQ-v2. Color of each point denotes its value learned by $Q(s, a)$ or $\overline{Q}(\overline{s}, \overline{a})$. Points are also projected onto each main plane. The homomorphism map of DHPG has mapped the latent states of corresponding legs (e.g., left forward leg and right backward leg) **(a)** on to the same abstract latent states **(b)**, indicating a clear structure in $\overline{S}$.

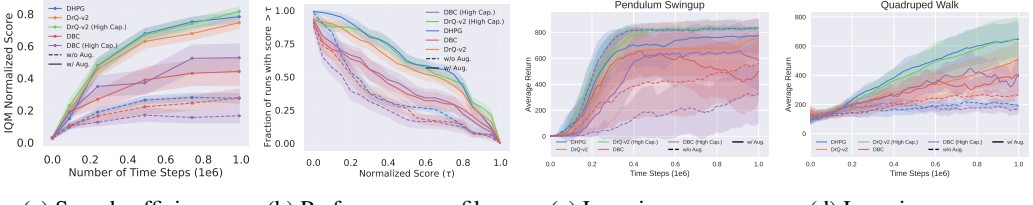

| (a) Sample efficiency. | (b) Performance profiles. | (c) Learning curves. | (d) Learning curves. |

Figure 7: Results of DM Control tasks with **pixel observations** for **higher-capacity variants** of DBC and DrQ-v2 obtained on 10 seeds. RLiable metrics are aggregated over 14 tasks. **(a)** RLiable IQM scores as a function of number of steps for comparing sample efficiency, **(b)** RLiable performance profiles at 500k steps, **(c)**-**(d)** examples of learning curves. Full results are in Appendix D.7. Shaded regions represent 95% confidence intervals.

embedding for continuous control is learned. In contrast, we use the notion of homomorphisms to learn the state-dependent action representations, while preserving values. Action representations can be combined with temporal abstraction [77] for discovering extended actions [66, 3, 16, 17].

**State Representation Learning.** Extant methods for learning the underlying state space from raw observations often use latent models [29, 36, 37, 34, 10], auxiliary prediction tasks [44, 53, 54], physics-inspired inductive biases [45, 19, 31], unsupervised learning [42, 52], or self-supervised learning [6, 71, 38, 39, 24]. From another point of view, representation learning can be effectively decoupled from the RL problem [23, 74]. Symmetries of the environment can also be used for representation learning [58, 55, 61, 84, 40, 41, 62, 13]. In fact, MDP homomorphisms are specializations of such approaches for RL. A key distinguishing factor of MDP homomorphisms is their ability to take actions into account for representation learning in the same premises as Thomas et al. [80]. Recently, simple image augmentation methods have shown significant improvements in RL performance [89, 48]. Since these approaches are in general orthogonal to state abstractions, they can be combined together.

## 9   Conclusion

In this paper, we developed the novel theory of continuous MDP homomorphisms using measure theory, and we rigorously proved their value and optimal value equivalence properties. We derived the homomorphic PG in order to directly use a joint state-action abstraction for policy optimization. Importantly, we rigorously proved that applying our homomorphic PG on the abstract MDP is equivalent to applying the standard DPG on the actual MDP. Based on our novel theoretical results, we developed a deep actor-critic algorithm that can simultaneously learn the policy and the MDP homomorphism map using the lax bisimulation metric. Our algorithm improves upon strong baselines in both learning from state and pixel observations. The visualization of the latent space demonstrates the strong potential of MDP homomorphisms in learning structured representations that can preserve value functions. We believe that our work will open-up future possibilities for the application of MDP homomorphisms in challenging continuous control problems.

## Acknowledgements

SRS is supported by an NSERC CGS-D scholarship. RZ was supported by an NSERC CGS-M scholarship at the time this work was completed. We would like to thank Juan Camilo Gamboa Higuera, Harley Wiltzer, and Scott Fujimoto for insightful discussions. The computing resources for this research were provided by Calcul Quebec and the Digital Research Alliance of Canada.

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
