# Supplementary Material: Continuous MDP Homomorphisms and Homomorphic Policy Gradient

**Sahand Rezaei-Shoshtari**
McGill University and Mila

**Rosie Zhao**
McGill University and Mila

**Prakash Panangaden**
McGill University and Mila

**David Meger**
McGill University and Mila

**Doina Precup**
McGill University, Mila, and DeepMind

## A  Additional Background

### A.1  Background on the Policy Gradient Theorem

RL algorithms can be broadly divided into *value-based* and *policy gradient* (PG) methods. While value-based methods select actions via a greedy maximization step based on the learned action-values, PG methods directly optimize a parameterized policy $\pi_\theta$ based on the performance gradient $\nabla_\theta J(\theta)$. Thus, unlike value-based methods, PG algorithms inherit the strong, albeit local, convergence guarantees of the gradient descent and are naturally extendable to continuous actions. The fundamental theorem underlying PG methods is the *policy gradient theorem* [10]:

$$\nabla_\theta J(\pi_\theta) = \int_{s \in \mathcal{S}} \rho^{\pi_\theta}(s) \int_{a \in \mathcal{A}} \nabla_\theta \pi_\theta(a|s) Q^{\pi_\theta}(s, a) \tag{1}$$

where $\rho^{\pi_\theta}(s) = \lim_{t \to \infty} \gamma^t P(s_t = s | s_0, a_{0:t} \sim \pi_\theta)$ is the discounted stationary distribution of states under $\pi_\theta$ which is assumed to exist and to be independent of the initial state distribution (ergodicity assumption). The significance of the PG theorem is that the effect of policy changes on the state distribution does not appear in its expression, allowing for a sample-based estimate of the gradient [13].

The deterministic policy gradient (DPG) is derived for deterministic policies by Silver et al. [8] as:

$$\nabla_\theta J(\pi_\theta) = \int_{s \in \mathcal{S}} \rho^{\pi_\theta}(s) \nabla_\theta \pi_\theta(s) \nabla_a Q^{\pi_\theta}(s, a) \big|_{a = \pi_\theta(s)} \tag{2}$$

Since DPG does not need to integrate over the action space, it is often more sample-efficient than the stochastic policy gradient [8]. However, a noise needs to be manually injected during exploration as the deterministic policy does not have any inherent means of exploration. Finally, it is worth noting that due to the differentiation of the value function with respect to $a$, DPG is only applicable to continuous actions.

### A.2  Mathematical Tools

Various mathematical concepts from measure theory and differential geometry are presented in this section. We only explicitly introduce concepts which are directly mentioned or relevant to the proofs presented in section C; for a more comprehensive overview, we direct the reader to textbooks such as [3, 5, 9].

**Definition 1** ($\sigma$-algebra). Given a set $X$, a $\sigma$-algebra on $X$ is a family $\Sigma$ of subsets of $X$ such that 1) $X \in \Sigma$, 2) $A \in \Sigma$ implies $A^c \in \Sigma$ (closure under complements), and 3) if $(A_i)_{i \in \mathbb{N}}$ satisfies $A_i \in \Sigma$ for all $i \in \mathbb{N}$, then $\cup_{i \in \mathbb{N}} A_i \in \Sigma$ (closure under countable union). The tuple $(X, \Sigma)$ is a measurable space.

The $\sigma$-algebra of a space specifies the sets in which a measure is defined; in probability theory— and in our use case— a $\sigma$-algebra represents a collection of events which can be assigned probabilities.

36th Conference on Neural Information Processing Systems (NeurIPS 2022).

**Definition 2** (Pushforward measure). Let $(X_1, \Sigma_1)$ and $(X_2, \Sigma_2)$ be two measurable spaces, $f : X_1 \to X_2$ a measurable map and $\mu : \Sigma_1 \to [0, \infty]$ a measure on $X_1$. Then the pushforward measure of $\mu$ with respect to $f$, denoted $f_*(\mu) : \Sigma_2 \to [0, \infty]$ is defined as:

$$(f_*(\mu))(B) = \mu(f^{-1}(B)) \ \forall \ B \in \Sigma_2.$$

**Theorem 1** (Change of variables). *A measurable function $g$ on $X_2$ is integrable with respect to $f_*(\mu)$ if and only if the function $g \circ f$ is integrable with respect to $\mu$, in which case the integrals are equal:*

$$\int_{X_2} g d(f_*(\mu)) = \int_{X_1} g \circ f d\mu.$$

**Definition 3** (Local diffeomorphism). Let $M$ and $N$ be differentiable manifolds. A function $f : M \to N$ is a *local diffeomorphism*, if for each point $x \in M$ there exists an open set $U$ containing $x$ such that $f(U)$ is open in $N$ and $f|_U : U \to f(U)$ is a diffeomorphism.

**Theorem 2** (Inverse function theorem for manifolds). *If $f : M \to N$ is a smooth map whose differential $df_x : T_x M \to T_{f(x)} N$ is an isomorphism at a point $x \in M$. Then $f$ is a local diffeomorphism at $x$.*

**Theorem 3** (Chain rule for manifolds). *If $f : M \to N$ and $g : N \to O$ are smooth maps of manifolds, then:*

$$d(g \circ f)_x = dg_{f(x)} \circ df_x.$$

# B    Assumptions and Conditions

The derivation of our homomorphic policy gradient theorem is for continuous state and action spaces. Therefore, we have assumed the following regularity conditions on the actual MDP $\mathcal{M}$ and its MDP homomorphic image $\overline{\mathcal{M}}$ under the MDP homomorphism map $h$. The conditions are largely based on the regularity conditions of the deterministic policy gradient theorem [8]:

**Regularity conditions 1:** $\tau_a(s'|s)$, $\nabla_a \tau_a(s'|s)$, $\overline{\tau}_{\overline{a}}(\overline{s}'|\overline{s})$, $\nabla_{\overline{a}} \overline{\tau}_{\overline{a}}(\overline{s}'|\overline{s})$, $R(s,a), \nabla_a R(s,a)$, $\overline{R}(\overline{s}, \overline{a}), \nabla_{\overline{a}} \overline{R}(\overline{s}, \overline{a}), \pi_\theta^\uparrow(s), \nabla_\theta \pi_\theta^\uparrow(s), \overline{\pi}_\theta(\overline{s}), \nabla_\theta \overline{\pi}_\theta(\overline{s}), p_1(s)$, and $\overline{p}_1(\overline{s})$ are continuous with respect to all parameters and variables $s, \overline{s}, a, \overline{a}, s'$, and $\overline{s}'$.

**Regularity conditions 2:** There exists a $b$ and $L$ such that $\sup_s p_1(s) < b$, $\sup_{\overline{s}} \overline{p}_1(\overline{s}) < b$, $\sup_{a,s,s'} \tau_a(s'|s) < b$, $\sup_{\overline{a}, \overline{s}, \overline{s}'} \overline{\tau}_{\overline{a}}(\overline{s}'|\overline{s}) < b$, $\sup_{a,s} R(s,a) < b$, $\sup_{\overline{a}, \overline{s}} \overline{R}(\overline{s}, \overline{a}) < b$, $\sup_{a,s,s'} \|\nabla_a \tau_a(s'|s)\| < L, \sup_{\overline{a}, \overline{s}, \overline{s}'} \|\nabla_{\overline{a}} \overline{\tau}_{\overline{a}}(\overline{s}'|\overline{s})\| < L$, $\sup_{s,a} \|\nabla_a R(s,a)\| < L, \sup_{\overline{s}, \overline{a}} \|\nabla_{\overline{a}} \overline{R}(\overline{s}, \overline{a})\| < L$.

We also assume the following conditions on the continuous MDP homomorphism map $h = (f, g_s)$, as discussed in Definition 3:

**Regularity conditions 3:** The action mapping $g_s(a)$ is a local diffeomorphism (Definition 3). Hence it is continuous with respect to $a$ and locally bijective with respect to $a$. Additionally, $\nabla_a g_s(a)$ is continuous with respect to the parameter $a$, and there exists a $L$ such that $\sup_{s,a} \|\nabla_a g_s(a)\| < L$.

## C Proofs

Below are the proofs accompanying Sections 3, 4 and 5.

### C.1 Proof of Theorem 1: Value Equivalence

*Proof.* The proof is along the lines of the *optimal value equivalence* theorem of Ravindran and Barto [7]. We define the $m$-step discounted action value function $Q_m^{\pi^\uparrow}(s, a)$ recursively for all $(s, a) \in \mathcal{S} \times \mathcal{A}$ and for all integers $m \geq 1$ as:

$$Q_m^{\pi^\uparrow}(s, a) = R(s, a) + \gamma \sum_{s' \in \mathcal{S}} \tau_a(s'|s) \sum_{a' \in \mathcal{A}} \pi^\uparrow(a'|s') Q_{m-1}^{\pi^\uparrow}(s', a'),$$

with $Q_0^{\pi^\uparrow}(s, a) = R(s, a)$. The proof is by induction on $m$; the base case of $m = 0$ is true because:

$$Q_0^{\pi^\uparrow}(s, a) = R(s, a) = \overline{R}(f(s), g_s(a)) = Q_0^{\overline{\pi}}(f(s), g_s(a)).$$

Now suppose towards induction that $Q_k^{\pi^\uparrow}(s, a) = Q_k^{\overline{\pi}}(f(s), g_s(a))$ for all values of $k$ less than $m$ and all state action pairs $(s, a) \in \mathcal{S} \times \mathcal{A}$. Using the fact that $h = (f, g_s)$ is an MDP homomorphism, we have:

$$Q_m^{\pi^\uparrow}(s, a) = R(s, a) + \gamma \sum_{s' \in \mathcal{S}} \tau_a(s'|s) \sum_{a' \in \mathcal{A}} \pi^\uparrow(a'|s') Q_{m-1}^{\pi^\uparrow}(s', a')$$

$$= R(s, a) + \gamma \sum_{[s']_{B_h|\mathcal{S}} \in B_h|\mathcal{S}} \sum_{s'' \in [s']_{B_h|\mathcal{S}}} \tau_a(s''|s) \sum_{a' \in \mathcal{A}} \pi^\uparrow(a'|s') Q_{m-1}^{\overline{\pi}}(f(s'), g_{s'}(a')) \quad (3)$$

$$= R(s, a) + \gamma \sum_{[s']_{B_h|\mathcal{S}} \in B_h|\mathcal{S}} \sum_{s'' \in [s']_{B_h|\mathcal{S}}} \tau_a(s''|s) \sum_{\overline{a}' \in \overline{\mathcal{A}}} \sum_{a'' \in g_{s'}^{-1}(\overline{a}')} \pi^\uparrow(a''|s') Q_{m-1}^{\overline{\pi}}(f(s'), \overline{a}')$$

$$= \overline{R}(f(s), g_s(a)) + \gamma \sum_{[s']_{B_h|\mathcal{S}} \in B_h|\mathcal{S}} \overline{\tau}_{g_s(a)}(f(s')|f(s)) \sum_{\overline{a}' \in \overline{\mathcal{A}}} \overline{\pi}(\overline{a}'|f(s')) Q_{m-1}^{\overline{\pi}}(f(s'), \overline{a}')$$

$$(4)$$

$$= \overline{R}(f(s), g_s(a)) + \gamma \sum_{\overline{s}' \in \overline{\mathcal{S}}} \overline{\tau}_{g_s(a)}(\overline{s}'|f(s)) \sum_{\overline{a}' \in \overline{\mathcal{A}}} \overline{\pi}(\overline{a}'|\overline{s}') Q_{m-1}^{\overline{\pi}}(\overline{s}', \overline{a}')$$

$$= Q_m^{\overline{\pi}}(f(s), g_s(a)).$$

Where in equation (3) we used the fact that $Q_{m-1}^{\pi^\uparrow}(s, a) = Q_{m-1}^{\overline{\pi}}(f(s), g_s(a))$ from the induction assumption. In equation (4) we used $\sum_{s'' \in [s']_{B_h|\mathcal{S}}} \tau_a(s''|s) = \overline{\tau}_{g_s}(f(s')|f(s))$ and $\sum_{a'' \in g_{s'}^{-1}(\overline{a}')} \pi^\uparrow(a''|s') = \overline{\pi}(\overline{a}'|f(s'))$ from the definition of MDP homomorphism [7]. Since $R$ and $\overline{R}$ are bounded, it follows by induction that $Q^{\pi^\uparrow}(s, a) = Q^{\overline{\pi}}(f(s), g_s(a))$ for all $(s, a) \in \mathcal{S} \times \mathcal{A}$.

The proof for $V^{\pi^\uparrow}(s) = V^{\overline{\pi}}(f(s))$ follows directly from the equivalence of action value functions and the fact that the two policies are tied together through the lifting process because in general we have: $V^\pi(s) = \sum_{a \in A} \pi(a|s) Q^\pi(s, a)$. $\qquad \square$

### C.2 Proof of Theorem 2: Optimal Value Equivalence for Continuous MDP Homomorphisms

*Proof.* The proof follows along the same lines as Ravindran and Barto [7]. We will first prove the following claim:

**Claim.** For $m \geq 1$, define the sequence $Q_m : \mathcal{S} \times \mathcal{A} \to \mathbb{R}$ as

$$Q_m(s, a) = R(s, a) + \gamma \int_{s' \in \mathcal{S}} \tau_a(ds'|s) \sup_{a' \in \mathcal{A}} Q_{m-1}(s', a')$$

and $Q_0(s, a) = R(s, a)$. Define the sequence $\overline{Q}_m : \overline{\mathcal{S}} \times \overline{\mathcal{A}} \to \mathbb{R}$ analogously. Then for any $(s, a) \in \mathcal{S} \times \mathcal{A}$ we have

$$Q_m(s, a) = \overline{Q}_m(f(s), g_s(a)).$$

We will prove this claim by induction on $m$. The base case $m = 0$ follows from the reward invariance property of continuous MDP homomorphisms:

$$Q_0(s,a) = R(s,a) = \overline{R}(f(s), g_s(a)) = \overline{Q}_0(f(s), g_s(a)).$$

For the inductive case, note that

$$Q_m(s,a) = R(s,a) + \gamma \int_{s' \in \mathcal{S}} \tau_a(ds'|s) \sup_{a' \in \mathcal{A}} Q_{m-1}(s', a')$$

$$= \overline{R}(f(s), g_s(a)) + \gamma \int_{s' \in \mathcal{S}} \tau_a(ds'|s) \sup_{a' \in \mathcal{A}} \overline{Q}_{m-1}(f(s'), g_{s'}(a')) \tag{5}$$

$$= \overline{R}(f(s), g_s(a)) + \gamma \int_{s' \in \mathcal{S}} \tau_a(ds'|s) \sup_{\overline{a}' \in \overline{\mathcal{A}}} \overline{Q}_{m-1}(f(s'), \overline{a}') \tag{6}$$

$$= \overline{R}(f(s), g_s(a)) + \gamma \int_{\overline{s}' \in \overline{\mathcal{S}}} \overline{\tau}_{g_s(a)}(d\overline{s}'|f(s)) \sup_{\overline{a}' \in \overline{\mathcal{A}}} \overline{Q}_{m-1}(\overline{s}', \overline{a}') \tag{7}$$

$$= Q_{m-1}(f(s), g_s(a)), \tag{8}$$

where Equation 5 follows from the inductive hypothesis, Equation 6 follows from $g_s$ being surjective, and Equation 7 follows from the change of variables formula (Theorem 1); indeed, from Definition 3 we have the pushforward measure of $\tau_a(\cdot|s)$ with respect to $f$ equals $\tau_{g_s(a)}(\cdot|f(s))$ and here $g : \overline{\mathcal{S}} \to \mathbb{R}$ is defined as $g(\overline{s}) = \sup_{\overline{a}' \in \overline{\mathcal{A}}} \overline{Q}_{m-1}(\overline{s}, \overline{a}')$. This concludes the induction proof. Since $\lim_{m \to \infty} Q_m(s,a) = Q^*(s,a)$, it follows that $Q^*(s,a) = \overline{Q}^*(f(s), g_s(a))$.

The proof for $V^*(s) = \overline{V}^*(f(s))$ follows directly from the equivalence of optimal action value functions as $V^*(s) = \max_a Q^*(s,a)$ in general. $\qquad\square$

### C.3 Proof of Theorem 3: Value Equivalence for Deterministic Policies and Continuous MDP Homomorphisms

*Proof.* Unlike the proofs of Theorems 1 and 2, here we assume the policy is deterministic due to the complications of lifting stochastic policies discussed in Section 4.2. Therefore, the lifting process can be simply obtained as $\pi^\uparrow(s) = g_s^{-1}(\overline{\pi}(f(s)))$ and the inverse of the lifting process is $\overline{\pi}(f(s)) = g_s(\pi^\uparrow(s))$, as the mapping $g_s$ is assumed to be an invertible continuous map (Appendix B).

Similarly to Ravindran and Barto [7], the proof is by induction. We define the $m$-step discounted action value function $Q_m^{\pi^\uparrow}(s,a)$ for the domain $\mathcal{S} \times \mathcal{A}$ and for all integers $m \geq$ as:

$$Q_m^{\pi^\uparrow}(s,a) = R(s,a) + \gamma \int_{s' \in \mathcal{S}} \tau_a(ds'|s) Q_{m-1}^{\pi^\uparrow}(s', \pi^\uparrow(s')),$$

with $Q_0^{\pi^\uparrow}(s,a) = R(s,a)$ for all pairs $(s,a) \in \mathcal{S} \times \mathcal{A}$. The proof is by induction on $m$, the base case of $m = 0$ is true because:

$$Q_0^{\pi^\uparrow}(s,a) = R(s,a) = \overline{R}(f(s), g_s(a)) = Q_0^{\overline{\pi}}(f(s), g_s(a)).$$

Now suppose towards induction that $Q_k^{\pi^\uparrow}(s,a) = Q_k^{\overline{\pi}}(f(s), g_s(a))$ for all values of $k$ less than $m$ on the domain $\mathcal{S} \times \mathcal{A}$. Using the fact that $h = (f, g_s)$ is a continuous MDP homomorphism, we have:

$$Q_m^{\pi^\uparrow}(s,a) = R(s,a) + \gamma \int_{s' \in \mathcal{S}} \tau(ds'|s) Q_{m-1}^{\pi^\uparrow}(s', \pi^\uparrow(s'))$$

$$= R(s,a) + \gamma \int_{s' \in \mathcal{S}} \tau_a(ds'|s) Q_{m-1}^{\overline{\pi}}(f(s'), g_{s'}(\pi^\uparrow(s'))) \tag{9}$$

$$= \overline{R}(f(s), g_s(a)) + \gamma \int_{s' \in \mathcal{S}} \tau_a(ds'|s) Q_{m-1}^{\overline{\pi}}(f(s'), \overline{\pi}(f(s'))) \tag{10}$$

$$= \overline{R}(f(s), g_s(a)) + \gamma \int_{\overline{s} \in \overline{\mathcal{S}}} \overline{\tau}_{g_s(a)}(d\overline{s}|f(s)) Q_{m-1}^{\overline{\pi}}(\overline{s}', \overline{\pi}(\overline{s}')) \tag{11}$$

$$= Q_m^{\overline{\pi}}(f(s), g_s(a)). \tag{12}$$

Where in equation (9), we used the induction assumption, in equation (10) we used the definition the inverse of policy lifting as defined above, and in equation (11) we applied the change of variables

formula (Theorem 1) using the fact that $\overline{\tau}_{g_s(a)}(\cdot|f(s))$ is the pushforward measure of $\tau_a(\cdot|s)$ under $f$ by definition. Since $R$ and $\overline{R}$ are bounded, it follows by induction that $Q^{\pi^\uparrow}(s,a) = Q^{\overline{\pi}}(f(s), g_s(a))$.

The proof for $V^{\pi^\uparrow}(s) = V^{\overline{\pi}}(f(s))$ follows directly from the equivalence of action value functions and the fact that the two policies are tied together through the lifting process because $V^\pi(s) = Q^\pi(s, \pi(s))$ for deterministic policies.

$\square$

## C.4 Proof of Theorem 4: Equivalence of Deterministic Policy Gradients

*Proof.* Assuming the conditions described in Appendix B, we first take the derivative of the deterministic policy lifting relation w.r.t. the policy parameters $\theta$ using the chain rule:

$$(g_s \circ \pi^\uparrow)(s) = (\overline{\pi} \circ f)(s)$$
$$d(g_s \circ \pi^\uparrow)_\theta(s) = d(\overline{\pi} \circ f)_\theta(s)$$
$$d(g_s)_{\pi^\uparrow(s)} \circ d(\pi^\uparrow)_\theta(s) = d(\overline{\pi} \circ f)_\theta(s)$$
$$\underbrace{\nabla_a g_s(a)\big|_{a=\pi^\uparrow(s)}}_{P} \nabla_\theta \pi^\uparrow(s) = \nabla_\theta \overline{\pi}(f(s)), \tag{13}$$

where $\circ$ is the composition operator and the dimensions of the matrices are $P \in \mathbb{R}^{|\overline{\mathcal{A}}| \times |A|}$, $\nabla_\theta \pi^\uparrow(s) \in \mathbb{R}^{|A| \times |\theta|}$, and $\nabla_\theta \overline{\pi}(\overline{s}) \in \mathbb{R}^{|\overline{\mathcal{A}}| \times |\theta|}$. Second, we take the derivative of the value equivalence theorem w.r.t. the actions $a$ using the chain rule:

$$Q^{\pi^\uparrow}(s,a) = Q^{\overline{\pi}}(f(s), g_s(a))$$
$$dQ^{\pi^\uparrow}(s,a)_a = dQ^{\overline{\pi}}(f(s), g_s(a))_a$$
$$\nabla_a Q^{\pi^\uparrow}(s,a)\big|_{a=\pi^\uparrow(s)} = \nabla_{\overline{a}} Q^{\overline{\pi}}(f(s), \overline{a})\big|_{\overline{a}=\overline{\pi}(f(s))} \underbrace{\nabla_a g_s(a)\big|_{a=g_s^{-1}(\overline{\pi}(f(s)))}}_{P}, \tag{14}$$

where the dimensions of the matrices are $\nabla_a Q^{\pi^\uparrow}(s,a) \in \mathbb{R}^{|A|}$, $\nabla_{\overline{a}} Q^{\overline{\pi}}(\overline{s}, \overline{a}) \in \mathbb{R}^{|\overline{\mathcal{A}}|}$, and similarly as before $P \in \mathbb{R}^{|\overline{\mathcal{A}}| \times |A|}$. As we assumed the $g_s$ to be a local diffeomorphism, the inverse function theorem (Theorem 2) states that the matrix $P$ is invertible, thus we right-multiply both sides of equation (14) by $P^{-1}$ and left-multiply the resulting equation by equation (13) to obtain the desired result:

$$\nabla_a Q^{\pi^\uparrow}(s,a)\big|_{a=\pi^\uparrow(s)} P^{-1} P \nabla_\theta \pi^\uparrow(s) = \nabla_{\overline{a}} Q^{\overline{\pi}}(f(s), \overline{a})\big|_{\overline{a}=\overline{\pi}(f(s))} \nabla_\theta \overline{\pi}(f(s))$$
$$\nabla_a Q^{\pi^\uparrow}(s,a)\big|_{a=\pi^\uparrow(s)} \nabla_\theta \pi^\uparrow(s) = \nabla_{\overline{a}} Q^{\overline{\pi}}(f(s), \overline{a})\big|_{\overline{a}=\overline{\pi}(f(s))} \nabla_\theta \overline{\pi}(f(s)). \tag{15}$$

$\square$

## C.5 Proof of Theorem 5: Homomorphic Policy Gradient

*Proof.* The proof follows along the same lines of the deterministic policy gradient theorem [8], but with additional steps for changing the integration space from $\mathcal{S}$ to $\overline{\mathcal{S}}$. First, we derive a recursive expression for $\nabla_\theta V^{\pi_\theta^\uparrow}(s)$ as:

$$\nabla_\theta V^{\pi_\theta^\uparrow}(s) = \nabla_\theta Q^{\pi_\theta^\uparrow}\Big(s, \pi_\theta^\uparrow(s)\Big)$$
$$= \nabla_\theta \Big[ R(s, \pi_\theta^\uparrow(s)) + \gamma \int_\mathcal{S} \tau_{\pi_\theta^\uparrow(s)}(s, ds') V^{\pi_\theta^\uparrow}(s') \Big]$$
$$= \nabla_\theta \pi_\theta^\uparrow(s) \nabla_a R(s, a)\big|_{a=\pi_\theta^\uparrow(s)}$$
$$\quad + \gamma \int_\mathcal{S} \Big[ \tau_{\pi_\theta^\uparrow(s)}(s, ds') \nabla_\theta V^{\pi_\theta^\uparrow}(s') + \nabla_\theta \pi_\theta^\uparrow(s) \nabla_a \tau_a(s, ds')\big|_{a=\pi_\theta^\uparrow(s)} V^{\pi_\theta^\uparrow}(s') \Big] \tag{16}$$
$$= \nabla_\theta \pi_\theta^\uparrow(s) \nabla_a \Big[ R(s, a) + \gamma \int_\mathcal{S} \tau_a(s, ds') V^{\pi_\theta^\uparrow}(s') \Big]\big|_{a=\pi_\theta^\uparrow(s)} + \gamma \int_\mathcal{S} \tau_{\pi_\theta^\uparrow(s)}(s, ds') \nabla_\theta V^{\pi_\theta^\uparrow}(s')$$

$$= \nabla_\theta \pi_\theta^\uparrow(s) \nabla_a Q^{\pi_\theta^\uparrow}(s,a)\Big|_{a=\pi_\theta^\uparrow(s)} + \gamma \int_{\mathcal{S}} \tau_{\pi_\theta^\uparrow(s)}(s,ds') \nabla_\theta V^{\overline{\pi}_\theta}(f(s')) \tag{17}$$

$$= \nabla_\theta \pi_\theta^\uparrow(s) \nabla_a Q^{\pi_\theta^\uparrow}(s,a)\Big|_{a=\pi_\theta^\uparrow(s)} + \gamma \int_{\overline{\mathcal{S}}} \overline{\tau}_{g_s(\pi_\theta^\uparrow(s))}(f(s),d\overline{s}') \nabla_\theta V^{\overline{\pi}_\theta}(f(s')) \tag{18}$$

$$= \nabla_\theta \overline{\pi}_\theta(f(s)) \nabla_{\overline{a}} Q^{\overline{\pi}_\theta}(f(s),\overline{a})\Big|_{\overline{a}=\overline{\pi}_\theta(f(s))} + \gamma \int_{\overline{\mathcal{S}}} \overline{\tau}_{\overline{\pi}_\theta(\overline{s})}(\overline{s},d\overline{s}') \nabla_\theta V^{\overline{\pi}_\theta}(\overline{s}') \tag{19}$$

$$= \nabla_\theta \overline{\pi}_\theta(f(s)) \nabla_{\overline{a}} Q^{\overline{\pi}_\theta}(f(s),\overline{a})\Big|_{\overline{a}=\overline{\pi}_\theta(f(s))} + \gamma \int_{\overline{\mathcal{S}}} p(\overline{s} \to \overline{s}', 1, \overline{\pi}_\theta) \nabla_\theta V^{\overline{\pi}_\theta}(\overline{s}') d\overline{s}'.$$

Where $p(\overline{s} \to \overline{s}', t, \overline{\pi}_\theta)$ is the probability of going from $\overline{s}$ to $\overline{s}'$ under the policy $\overline{\pi}_\theta(\overline{s})$ in $t$ time steps. In equation (16) we were able to apply the Leibniz integral rule to exchange the order of derivative and integration because of the regularity conditions on the continuity of the functions. In equation (17) we used the value equivalence property, and in equation (18) we used the change of variables formula based on the pushforward measure (1) of $\tau_a(s,.)$ with respect to $f$. Finally, in equation (19) we used the equivalence of policy gradients from Theorem 4. By recursively rolling out the formula above, we obtain:

$$\nabla_\theta V^{\pi_\theta^\uparrow}(s) = \nabla_\theta \overline{\pi}_\theta(f(s)) \nabla_{\overline{a}} Q^{\overline{\pi}_\theta}(f(s),\overline{a})\Big|_{\overline{a}=\overline{\pi}_\theta(f(s))}$$

$$+ \gamma \int_{\overline{\mathcal{S}}} p(\overline{s} \to \overline{s}', 1, \overline{\pi}_\theta) \nabla_\theta \overline{\pi}_\theta(f(s')) \nabla_{\overline{a}} Q^{\overline{\pi}_\theta}(f(s'),\overline{a})\Big|_{\overline{a}=\overline{\pi}_\theta(f(s'))} d\overline{s}'$$

$$+ \gamma^2 \int_{\overline{\mathcal{S}}} p(\overline{s} \to \overline{s}', 1, \overline{\pi}_\theta) \int_{\overline{\mathcal{S}}} p(\overline{s}' \to \overline{s}'', 1, \overline{\pi}_\theta) \nabla_\theta V^{\pi_\theta^\uparrow}(f(s'')) d\overline{s}'' d\overline{s}'$$

$$= \nabla_\theta \overline{\pi}_\theta(f(s)) \nabla_{\overline{a}} Q^{\overline{\pi}_\theta}(f(s),\overline{a})\Big|_{\overline{a}=\overline{\pi}_\theta(f(s))}$$

$$+ \gamma \int_{\overline{\mathcal{S}}} p(\overline{s} \to \overline{s}', 1, \overline{\pi}_\theta) \nabla_\theta \overline{\pi}_\theta(f(s')) \nabla_{\overline{a}} Q^{\overline{\pi}_\theta}(f(s'),\overline{a})\Big|_{\overline{a}=\overline{\pi}_\theta(f(s'))} d\overline{s}'$$

$$+ \gamma^2 \int_{\overline{\mathcal{S}}} p(\overline{s} \to \overline{s}'', 2, \overline{\pi}_\theta) \nabla_\theta V^{\overline{\pi}_\theta}(f(s'')) d\overline{s}'' \tag{20}$$

$$\vdots$$

$$= \int_{\overline{\mathcal{S}}} \sum_{t=0}^{\infty} \gamma^t p(\overline{s} \to \overline{s}', t, \overline{\pi}_\theta) \nabla_\theta \overline{\pi}_\theta(f(s)) \nabla_{\overline{a}} Q^{\overline{\pi}_\theta}(f(s),\overline{a})\Big|_{\overline{a}=\overline{\pi}_\theta(f(s))} d\overline{s}'. \tag{21}$$

Where in equation (20) we exchanged the order of integration using the Fubini's theorem that requires the boundedness of $\|\nabla_\theta V^{\overline{\pi}_\theta}(s)\|$ as described in the regularity conditions. Finally, we take the expectation of $\nabla_\theta V^{\pi_\theta^\uparrow}(s)$ over the initial state distribution:

$$\nabla_\theta J(\theta) = \nabla_\theta \int_{\mathcal{S}} p_1(s) V^{\pi_\theta^\uparrow}(s) ds$$

$$= \int_{\mathcal{S}} p_1(s) \nabla_\theta V^{\pi_\theta^\uparrow}(s) ds$$

$$= \int_{\mathcal{S}} p_1(s) \int_{\overline{\mathcal{S}}} \sum_{t=0}^{\infty} \gamma^t p(\overline{s} \to \overline{s}', t, \overline{\pi}_\theta) \nabla_\theta \overline{\pi}_\theta(f(s)) \nabla_{\overline{a}} Q^{\overline{\pi}_\theta}(f(s),\overline{a})\Big|_{\overline{a}=\overline{\pi}_\theta(f(s))} d\overline{s}' ds$$

$$= \int_{\overline{\mathcal{S}}} \overline{p}_1(\overline{s}) \int_{\overline{\mathcal{S}}} \sum_{t=0}^{\infty} \gamma^t p(\overline{s} \to \overline{s}', t, \overline{\pi}_\theta) \nabla_\theta \overline{\pi}_\theta(f(s)) \nabla_{\overline{a}} Q^{\overline{\pi}_\theta}(f(s),\overline{a})\Big|_{\overline{a}=\overline{\pi}_\theta(f(s))} d\overline{s}' d\overline{s} \tag{22}$$

$$= \int_{\overline{\mathcal{S}}} \rho^{\overline{\pi}_\theta}(\overline{s}) \nabla_\theta \overline{\pi}_\theta(\overline{s}) \nabla_{\overline{a}} Q^{\overline{\pi}_\theta}(\overline{s},\overline{a})\Big|_{\overline{a}=\overline{\pi}_\theta(\overline{s})} d\overline{s}. \tag{23}$$

Where $\rho^{\overline{\pi}_\theta}(\overline{s})$ is the discounted stationary distribution induced by the policy $\overline{\pi}_\theta$. In equation (22) we used the change of variable formula. Similar to the steps before, we have used the Leibniz integral rule to exchange the order of integration and derivative, used Fubini's theorem to exchange the order of integration. $\qquad\square$

# D Full Results

As discussed in Section 7, we evaluate DHPG on continuous control tasks from DM Control on state and pixel observations. Importantly, to reliably evaluate our algorithm against the baselines and to correctly capture the distribution of results, we follow the best practices proposed by Agarwal et al. [1] and report the interquartile mean (IQM) and performance profiles aggregated on all tasks over 10 random seeds. While our baseline results are obtained using the official code, when possible, some of the results may differ from the originally reported ones due to the difference in the seed numbers and our goal to present a faithful representation of the true performance distribution [1].

We use the official implementations of DrQv2, DBC, and SAC-AE, while we re-implement DeepMDP due to the unavailability of the official code; See Appendix E.3 for full details on the baselines.

## D.1 State Observations

Figure 1 shows full results obtained on 18 DeepMind Control Suite tasks with state observations to supplement results of Section 7.1. Domains that require excessive exploration and large number of time steps (e.g., acrobot, swimmer, and humanoid) are not included in this benchmark.

Figures 2 and 3 respectively show performance profiles and aggregate metrics [1] on 17 tasks; hopper hop is removed from RLiable evaluation as none of the algorithms have acquired reasonable performance in 1 million steps.

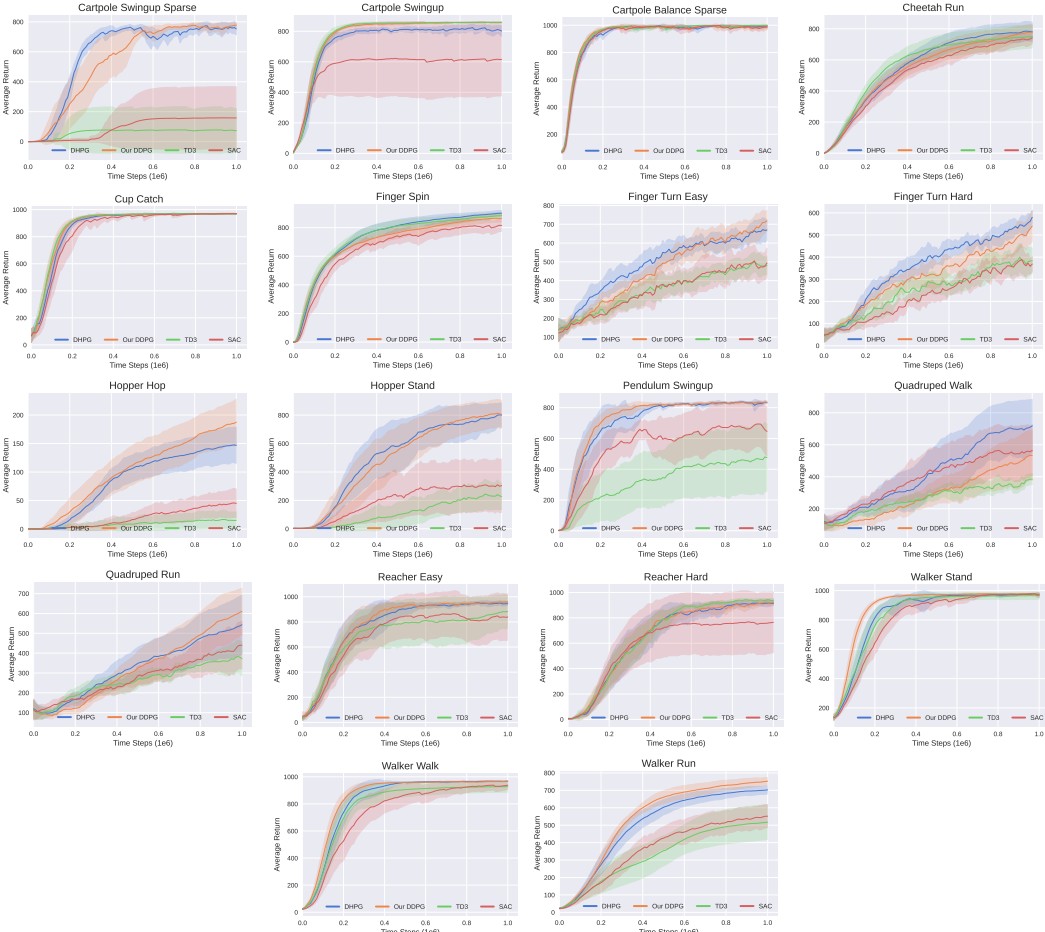

Figure 1: Learning curves for 18 DM control tasks with **state observations**. Mean performance is obtained over 10 seeds and shaded regions represent 95% confidence intervals. Plots are smoothed uniformly for visual clarity.

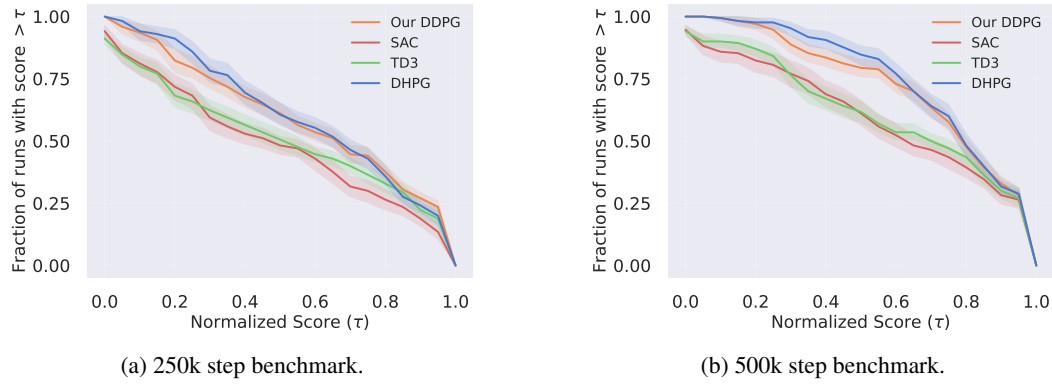

(a) 250k step benchmark.

(b) 500k step benchmark.

Figure 2: Performance profiles for **state observations** based on 17 tasks over 10 seeds, at 250k steps **(a)**, and at 500k steps **(b)**. Shaded regions represent $95\%$ confidence intervals.

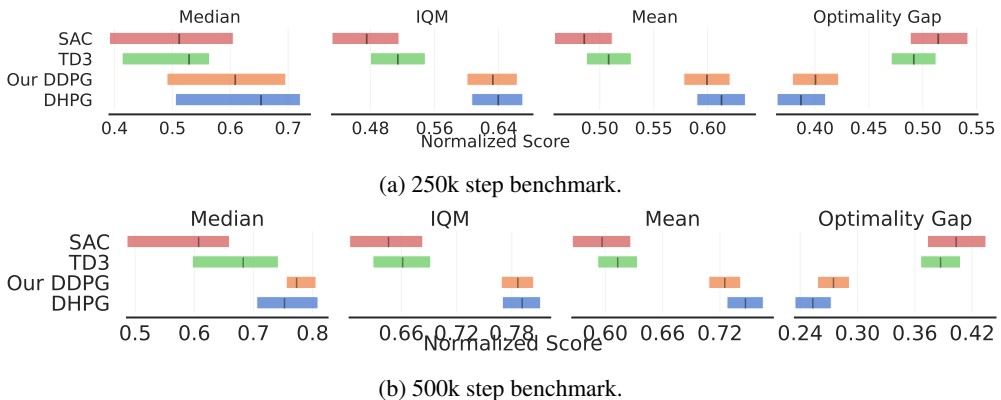

(a) 250k step benchmark.

(b) 500k step benchmark.

Figure 3: Aggregate metrics for **state observations** with $95\%$ confidence intervals based on 17 tasks over 10 seeds, at 250k steps **(a)**, and at 500k steps **(b)**.

## D.2 Pixel Observations

Figure 4 shows full results obtained on 16 DeepMind Control Suite tasks with pixel observations to supplement results of Section 7.2. Domains that require excessive exploration and large number of time steps (e.g., acrobot, swimmer, and humanoid) and domains with visually small targets (e.g., reacher hard and finger turn hard) are not included in this benchmark. In each plot, the solid lines present algorithms with image augmentation and dashed lines present algorithms without image augmentation.

Figures 5 and 6 respectively show performance profiles and aggregate metrics [1] on 14 tasks; hopper hop and walker run are removed from RLiable evaluation as none of the algorithms have acquired reasonable performance in 1 million steps.

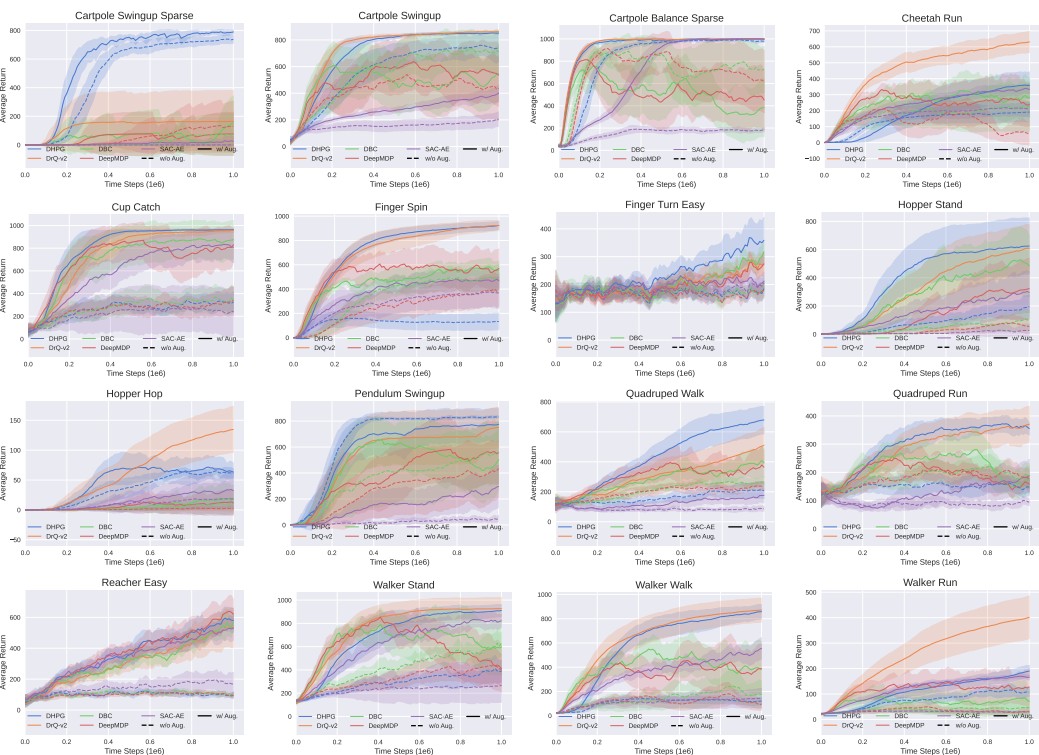

Figure 4: Learning curves for 16 DM control tasks with **pixel observations**. Mean performance is obtained over 10 seeds and shaded regions represent 95% confidence intervals. Plots are smoothed uniformly for visual clarity.

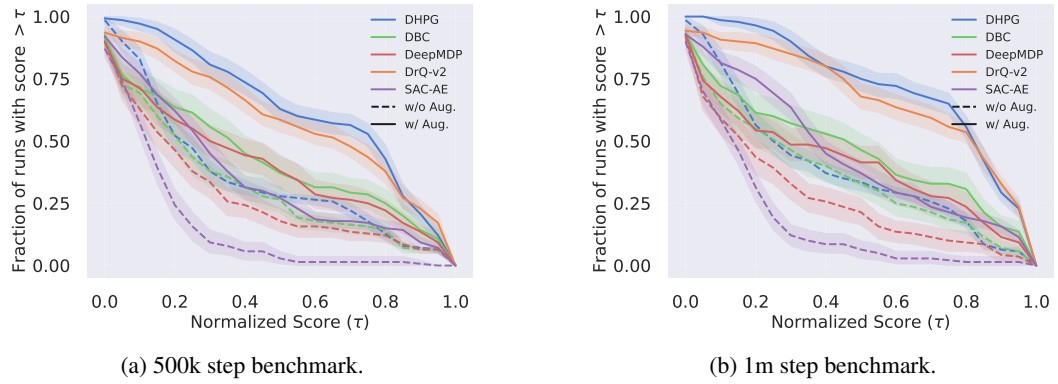

(a) 500k step benchmark.

(b) 1m step benchmark.

Figure 5: Performance profiles for **pixel observations** based on 14 tasks over 10 seeds, at 500k steps **(a)**, and at 1m steps **(b)**. Shaded regions represent $95\%$ confidence intervals.

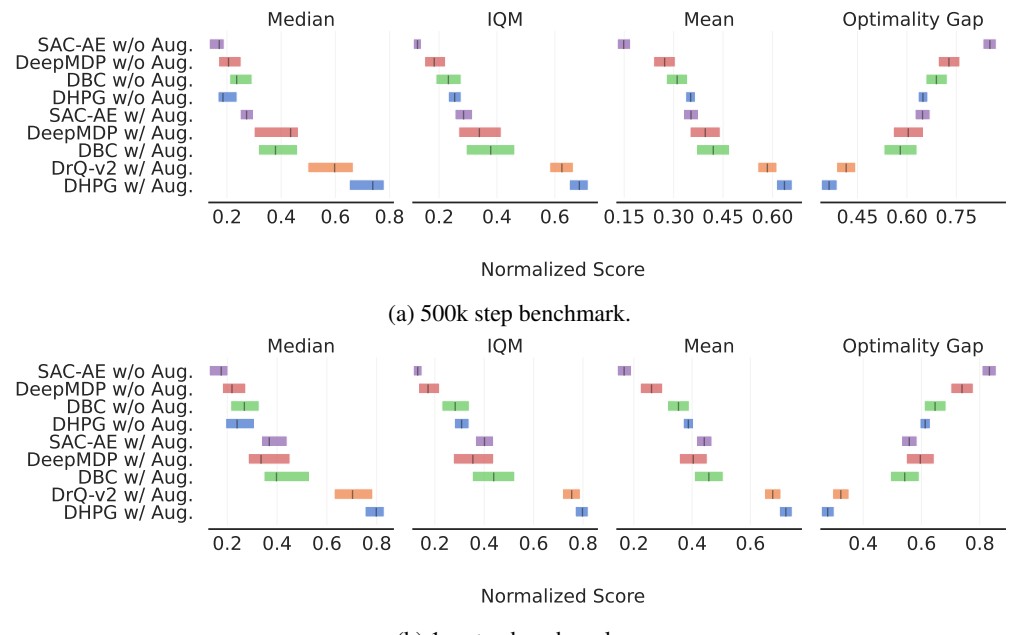

(a) 500k step benchmark.

(b) 1m step benchmark.

Figure 6: Aggregate metrics for **pixel observations** with $95\%$ confidence intervals based on 14 tasks over 10 seeds, at 500k steps **(a)**, and at 1m steps **(b)**.

### D.3 Transfer Learning Experiments

As discussed in Section 7.2, the purpose of transfer experiments is to ensure that using MDP homomorphisms does not compromise transfer abilities. Figure 7 shows learning curves for a series of transfer scenarios in which the critic, actor, and representations are transferred to a new task within the same domain. DHPG matches the same transfer abilities of other methods.

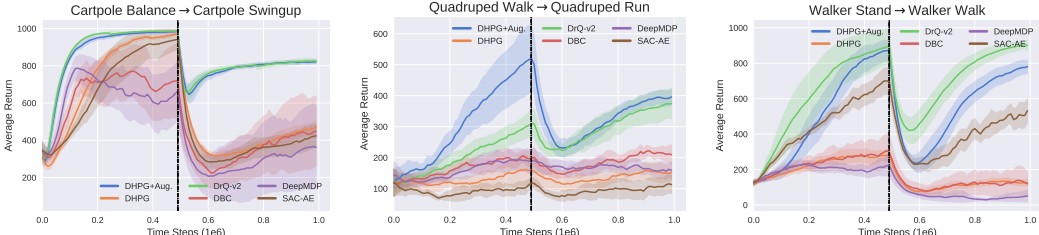

Figure 7: Learning curves for transfer experiments with **pixel observations**. At 500k time step mark, all components are transferred to a new task on the same domain. Mean performance is obtained over 10 seeds and shaded regions represent $95\%$ confidence intervals. Plots are smoothed uniformly for visual clarity.

### D.4 Value Equivalence Property in Practice

We can use the value equivalence between the critics of the actual and abstract MDPs as a measure for the quality of learned MDP homomorphismsm, since the two critics are not directly trained to minimize this distance, instead they have equivalent values through the learned MDP homomorphism map. Figure 8 shows the normalized mean absolute error of $|Q(s, a) - \overline{Q}(\overline{s}, \overline{a})|$ during training, indicating the property is holding in practice. Expectedly, for lower-dimensional tasks with easily learnable homomorphism maps (e.g., cartpole) the error is reduced earlier than more complicated tasks (e.g., quadruped and walker). But importantly, in all cases the error decreases over time and is at a reasonable range towards the end of the training, meaning the continuous MDP homomorphisms is adhering to conditions of Definition 3.

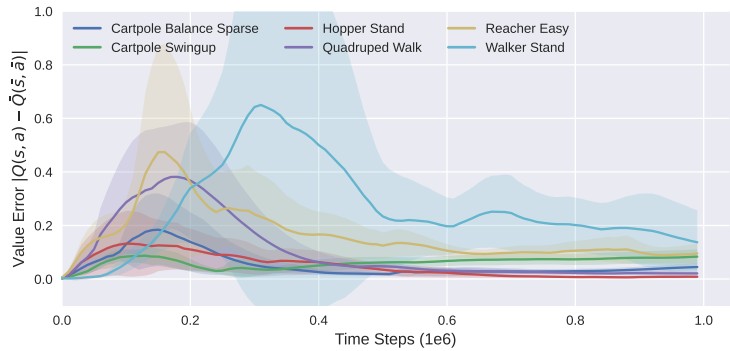

Figure 8: Normalized mean absolute error $|Q(s, a) - \overline{Q}(\overline{s}, \overline{a})|$ as a measure for the value equivalence property during training of different tasks from **pixel observations**. The error is measured on samples from the replay buffer and is normalzied by the range of the value function. The error is averaged over 10 seeds and shaded regions represent $95\%$ confidence intervals.

## D.5 Ablation Study on the Combination of HPG with DPG

We carry out an ablation study on the combination of HPG with DPG for actor updates as indicated discussed in Section 6. To that end, we evaluate the performance of four variants of DHPG (all using image augmentation) on pixel observations:

1. **DHPG:** Gradients of HPG and DPG are added together and a single actor update is done based on the sum of gradients. This is the standard DHPG algorithm that is used throughout the paper.
2. **DHPG with independent DPG update:** Gradients of HPG and DPG are independently used to update the actor.
3. **DHPG without DPG update:** Only HPG is used to update the actor.
4. **DHPG with single critic:** A single critic network is trained for learning values of both the actual and abstract MDP. Consequently, HPG and DPG are used to update the actor.

Figure 9 shows learning curves obtained on 16 DeepMind Control Suite tasks with pixel observations, and Figure 10 shows RLiable [1] evaluation metrics. In general, summing the gradients of HPG and DPG (variant 1) results in lower variance of gradient estimates compared to independent HPG and DPG updates (variant 2). Interestingly, the variant of DHPG without DPG (variant 3) performs reasonably well or even outperforms other variants in simple tasks where learning MDP homomorphisms is easy (e.g., cartpole and pendulum), indicating the effectiveness of our method in using **only** the abstract MDP to update the policy of the actual MDP. However, in the case of more complicated tasks (e.g., walker), DPG is required to additionally use the actual MDP for policy optimization. Finally, using a single critic for both the actual and abstract MDPs (variant 4) can improve sample efficiency in symmetrical MDPs, but may result in performance drops in non-symmetrical MDPs due to the large error bound between the two MDPs, $\|Q^{\pi^\uparrow}(s,a) - Q^{\overline{\pi}}(\overline{s}, \overline{a})\|$ [11].

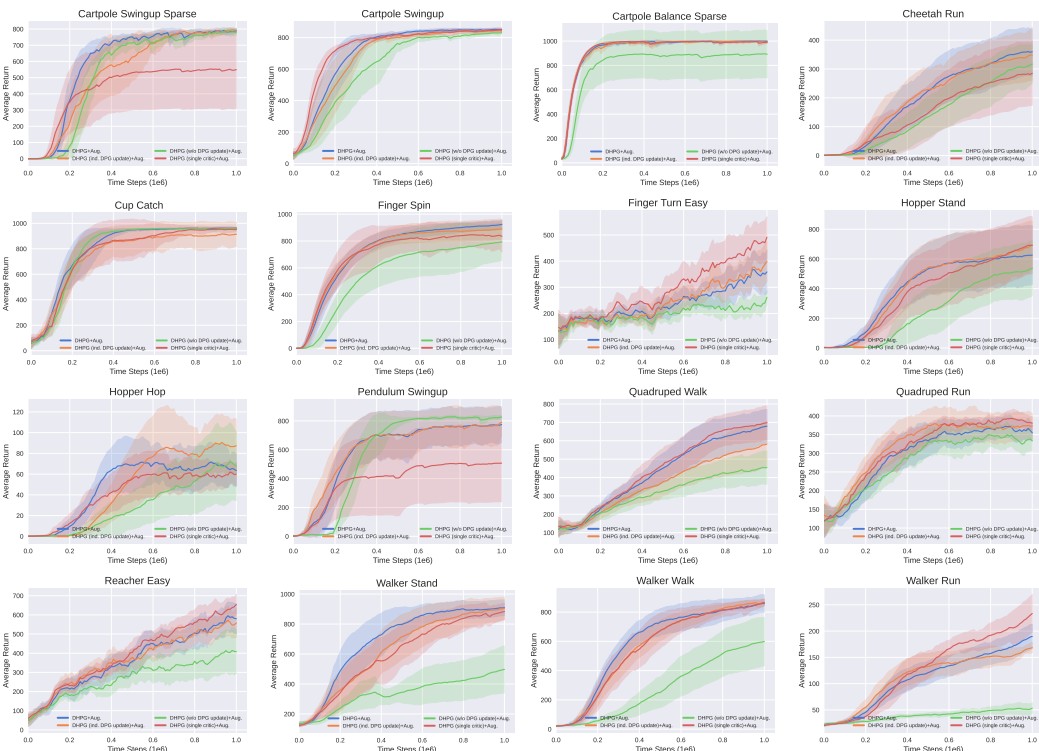

Figure 9: Ablation study on the combination of HPG and DPG. Learning curves for 16 DM control tasks with **pixel observations**. Mean performance is obtained over 10 seeds and shaded regions represent $95\%$ confidence intervals. Plots are smoothed uniformly for visual clarity.

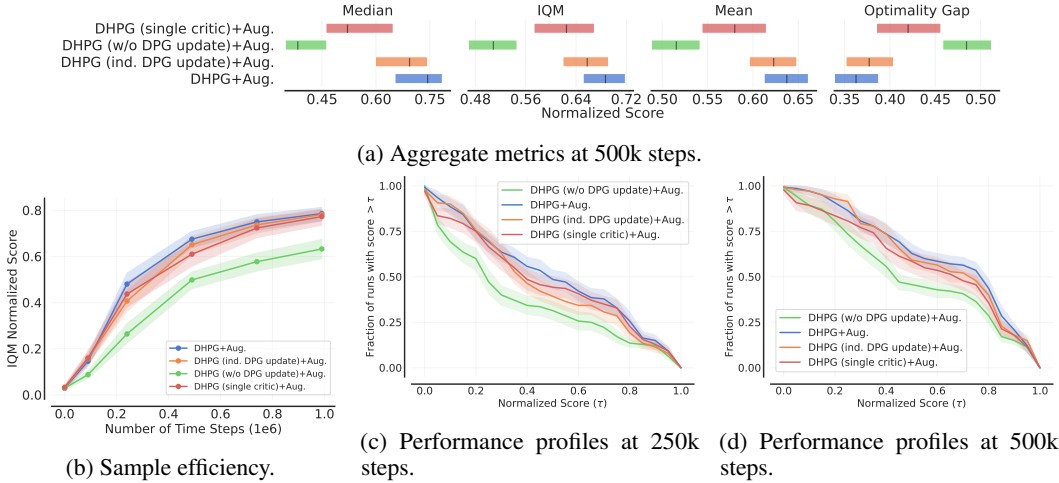

(a) Aggregate metrics at 500k steps.

(b) Sample efficiency.

(c) Performance profiles at 250k steps.

(d) Performance profiles at 500k steps.

Figure 10: Ablation study on the combination of HPG and DPG. RLiable evaluation metrics for **pixel observations** averaged on 14 tasks over 10 seeds. Aggregate metrics at 500k steps **(a)**, IQM scores as a function of number of steps for comparing sample efficiency **(b)**, performance profiles at 250k steps **(c)**, performance profiles at 500k steps **(d)**. Shaded regions represent $95\%$ confidence intervals.

### D.6 Ablation Study on n-step Return

We carry out an ablation study on the choice of $n$-step return for DHPG. Figure 11 shows RLiable [1] evaluation metrics for DHPG with 1-step and 3-step returns for pixel observations. We show the impact of $n$-step return on DHPG with and without image augmentation. Overall, $n$-step return appears to improve the early stages of training. In the case of DHPG without image augmentation, the final performance of 1-step return is better than 3-step return, perhaps indicating that using $n$-step return can render learning MDP homomorphisms more difficult.

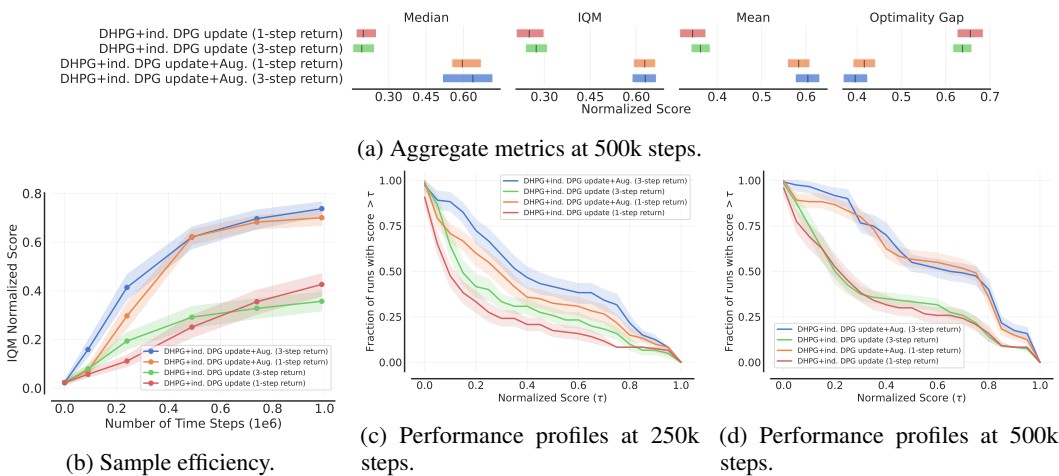

(a) Aggregate metrics at 500k steps.

(b) Sample efficiency.

(c) Performance profiles at 250k steps.

(d) Performance profiles at 500k steps.

Figure 11: Ablation study on $n$-step return. RLiable evaluation metrics for **pixel observations** averaged on 12 tasks over 10 seeds. Aggregate metrics at 1m steps **(a)**, IQM scores as a function of number of steps for comparing sample efficiency **(b)**, performance profiles at 250k steps **(c)**, and performance profiles at 500k steps **(d)**. Shaded regions represent $95\%$ confidence intervals.

## D.7 Comparison Against Higher-Capacity Baselines

The DHPG algorithm contains additional networks, such as the parameterized MDP homomorphism map and the abstract critic, thus it may have a higher network capacity compared to the baselines. To control for the effect of the network capacity and for a fair evaluation, we compare DHPG with higher-capacity variants of DBC and DrQ-v2 that have a larger critic networks. First, we provide a detailed list of network parameters based on the architecture described in Appendix E.2:

1. DHPG: image encoder (1,990,518) + actor (79,105) + critic (79,361) + dynamics model (117,348) + reward model (79,105) + abstract critic (91,905) + f (91,698) + g (91,954) = 2,620,994
2. DBC: image encoder (1,990,518) + actor (79,362) + critic (158,722) + dynamics model (104,804) + reward model (79,105) = 2,412,511
3. DrQ-v2: image encoder (1,990,518) + actor (79,105) + critic (158,722) = 2,228,602

To account for the parameter increase, we present variations of DBC and DrQ with a larger critic (512 hidden dim compared to the initial 256). Consequently, the new total number of parameters for DBC and DrQ are respectively 2,833,375 and 2,649,466. Figure 12 shows full results obtained on 16 DeepMind Control Suite tasks with pixel observations for higher-capacity variants of DBC and DrQ-v2 to supplement results of Section 7.2. Domains that require excessive exploration and large number of time steps (e.g., acrobot, swimmer, and humanoid) and domains with visually small targets (e.g., reacher hard and finger turn hard) are not included in this benchmark. In each plot, the solid lines present algorithms with image augmentation and dashed lines present algorithms without image augmentation.

Figures 13 and 14 respectively show performance profiles and aggregate metrics [1] on 14 tasks; hopper hop and walker run are removed from RLiable evaluation as none of the algorithms have acquired reasonable performance in 1 million steps.

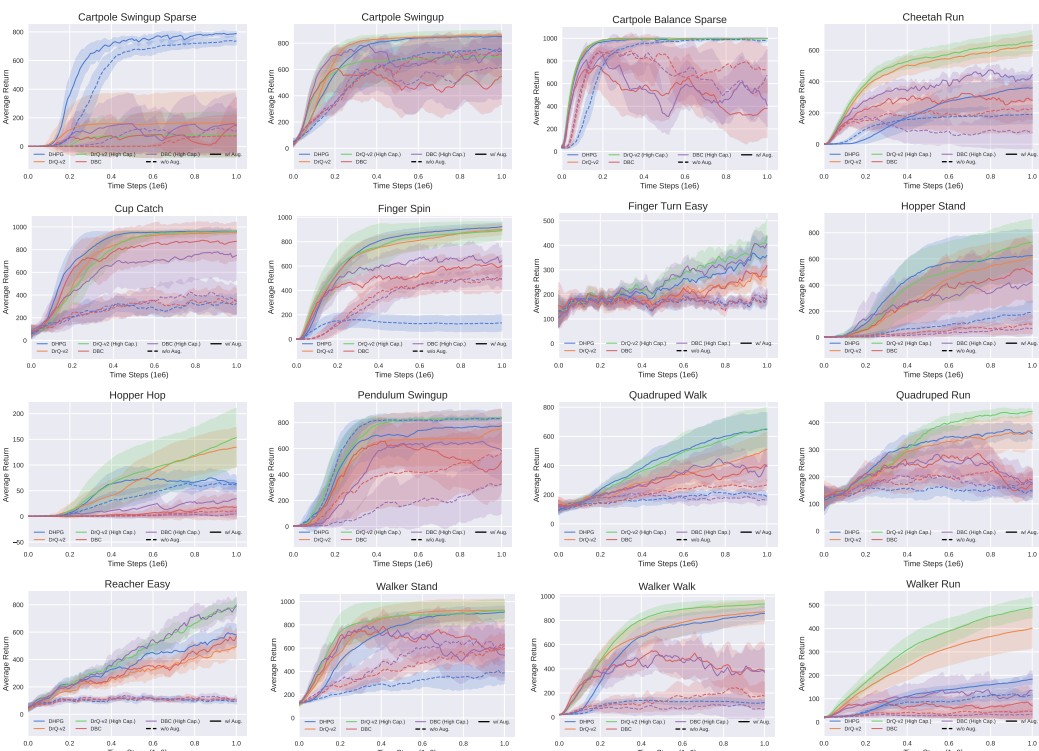

Figure 12: Learning curves for 16 DM control tasks with **pixel observations** for **higher-capacity variants** of DBC and DrQ-v2. Mean performance is obtained over 10 seeds and shaded regions represent 95% confidence intervals. Plots are smoothed uniformly for visual clarity.

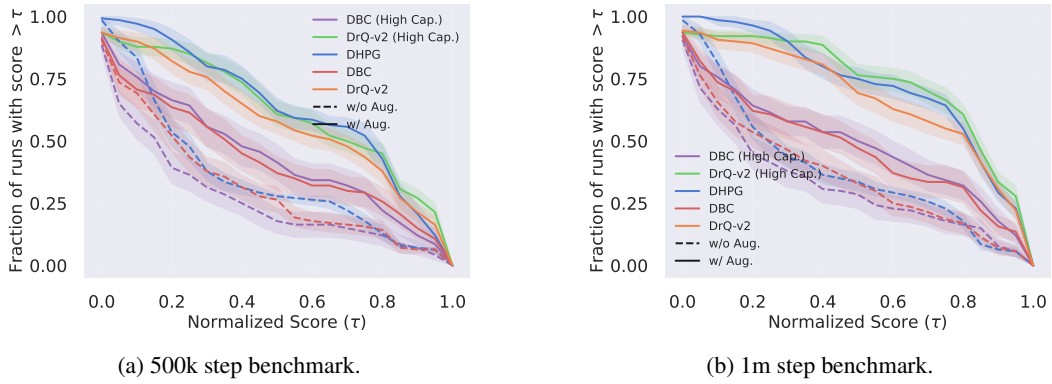

(a) 500k step benchmark.       (b) 1m step benchmark.

Figure 13: Performance profiles for **pixel observations** for **higher-capacity variants** of DBC and DrQ-v2 based on 14 tasks over 10 seeds, at 500k steps **(a)**, and at 1m steps **(b)**. Shaded regions represent 95% confidence intervals.

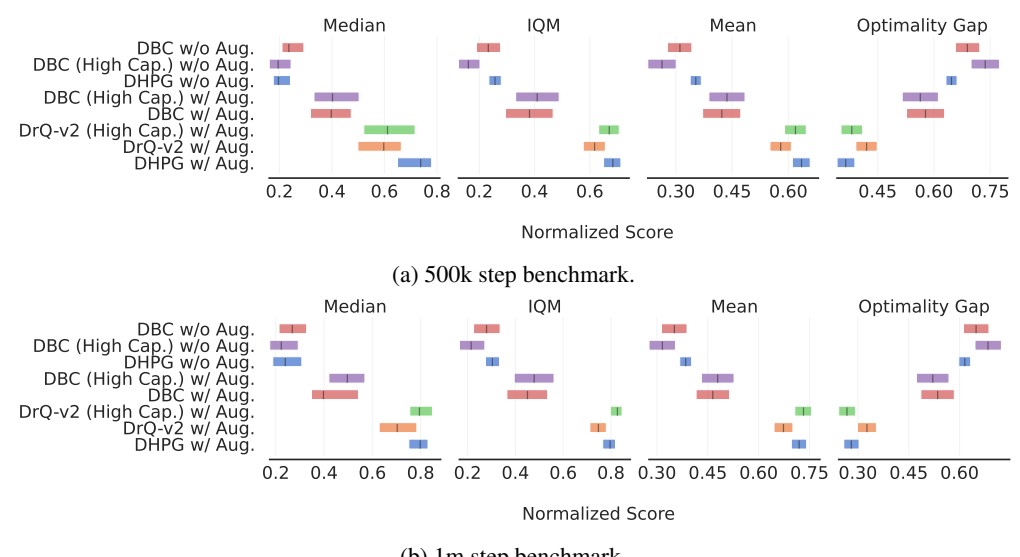

(a) 500k step benchmark.

(b) 1m step benchmark.

Figure 14: Aggregate metrics for **pixel observations** for **higher-capacity variants** of DBC and DrQ-v2 with 95% confidence intervals based on 14 tasks over 10 seeds, at 500k steps **(a)**, and at 1m steps **(b)**.

# E    Implementation Details

## E.1    Pseudo-code

Algorithm 1 presents the details of the Deep Homomorphic Policy Gradient (DHPG) for pixel observations. This is the main variant used throughout the paper, in which policy gradients obtained from DPG and HPG are added together before updating the actor. For clarity, here the TD error is estimated with 1-step returns.

In the image augmentation version of DHPG, as well as all the baselines, we use image augmentation of DrQ [15] that simply applies random shifts to pixel observations. First, $84 \times 84$ images are padded by 4 pixels (by repeating boundary pixels), and then a random $84 \times 84$ crop is selected, rendering the original image shifted by $\pm 4$ pixels. Similarly to Yarats et al. [14], we also apply bilinear interpolation on top of the shifted image by replacing each pixel value with the average of four nearest pixel values.

In order to use DHPG for state observations, Lines 8-11 should be simply removed.

---

**Algorithm 1** Deep Homomorphic Policy Gradient (DHPG) for Pixel Observations

---

1: **Hyperparameters:**
     Target network update weight $\alpha$, actor update delay $d$, clipped noise parameters $c$ and $\sigma$.
2: **Inputs:**
     Policy $\pi_\theta(s,a)$, actual critic $Q_\psi(s,a)$, abstract critic $\overline{Q}_{\overline{\psi}}(\overline{s},\overline{a})$, MDP homomorphism map
     $h_{\phi,\eta} = (f_\phi(s), g_\eta(s,a))$, reward predictor $\overline{R}_\rho(\overline{s})$, transition model $\tau_\nu(\overline{s}'|\overline{s},\overline{a})$, CNN image
     encoder $E_\mu$, and replay buffer $\mathcal{B}$.
3: Initialize target networks $\psi' \leftarrow \psi, \overline{\psi'} \leftarrow \overline{\psi}, \theta' \leftarrow \theta$.
4: **for** $t = 1$ **to** $T$ **do**
5:     Select action with exploration noise $a \sim \pi_\theta(E_\mu(s)) + \epsilon$, where $\epsilon \sim N(0,\sigma)$
6:     Store transition $(s,a,r,s')$ in $\mathcal{B}$
7:     Sample mini-batch $B_i \sim \mathcal{B}$

8:     **if** using image augmentation **then**
9:         $s \leftarrow \mathrm{aug}(s), \ s' \leftarrow \mathrm{aug}(s')$
10:    **end if**
11:    Encode pixel observations: $s \leftarrow E_\mu(s), \ s' \leftarrow E_\mu(s')$

12:    **Critic and MDP Homomorphism Update:**
13:    Compute MDP homomorphism loss: $\mathcal{L}_{\text{lax}}(\phi,\eta,\mu) + \mathcal{L}_{\text{h}}(\phi,\eta,\rho,\nu,\mu)$    ▷ Equations (12-13)
14:    Add clipped noise: $a' \leftarrow \pi_{\theta'}(s') + \epsilon$, where $\epsilon \sim \mathrm{clip}(N(0,\sigma),-c,c)$        ▷ TD3 [4]
15:    Compute critic loss: $\mathcal{L}_{\text{actual critic}}(\psi) + \mathcal{L}_{\text{abstract critic}}(\overline{\psi},\phi,\eta)$        ▷ Equations (9-10)
16:    Update: $\psi,\overline{\psi},\phi,\eta,\rho,\nu,\mu \leftarrow \arg\min_{\psi,\overline{\psi},\phi,\eta,\rho,\nu,\mu} \mathcal{L}_{\text{lax}} + \mathcal{L}_{\text{h}} + \mathcal{L}_{\text{actual critic}} + \mathcal{L}_{\text{abstract critic}}$
17:
18:    **Actor update:**
19:    **if** $t \mod d$ **then**
20:        Freeze $Q_\psi, \overline{Q}_{\overline{\psi}}, f_\phi, g_\eta$, and $E_\mu$
21:        Compute policy loss using DPG and HPG: $\mathcal{L}_{\text{actor}}(\theta)$                ▷ Equation (11)
22:        Update policy: $\theta \leftarrow \arg\min \mathcal{L}_{\text{actor}}(\theta)$
23:        Update target networks    $\psi' \leftarrow \alpha\psi + (1-\alpha)\psi', \ \overline{\psi}' \leftarrow \alpha\overline{\psi} + (1-\alpha)\overline{\psi}', \ \theta' \leftarrow \alpha\theta + (1-\alpha)\theta'$
24:    **end if**
25: **end for**

---

## E.2    Hyperparameters

Our code is submitted in the suplemental material.

We implemented our method in PyTorch [6] and results were obtained using Python v3.8.10, PyTorch v1.10.0, CUDA 11.4, and Mujoco 2.1.1 [12] on A100 GPUs on a cloud computing service. Tables 1-3 present the hyperparameters used in our experiments. The hyperparameters are all adapted from DrQ-v2 [14] *without any further hyperparameter tuning*. We have kept the same set of hyperparameters

across all algorithms and tasks, except for the walker domain which similarly to DrQ-v2 [14], we used $n$-step return of $n = 1$ and mini-batch size of 512.

The core RL components (actor and critic networks), as well as the components of DHPG (state and action encoders, transition and reward models) are all MLP networks with the ReLU activation function and one hidden layer with dimension of 256.

In the case of state observations, the abstract MDP has the same state and action dimensions as the actual MDP. In the case of pixel observations, the image encoder is based on the architecture of DrQ-v2 which is itself based on SAC-AE [16] and consists of four convolutional layers of $32 \times 3 \times 3$ with ReLU as their activation functions, followed by a one-layer fully-connected neural network with layer normalization [2] and tanh activation function. The stride of the convolutional layers are 1, except for the first layer which has stride 2. The image decoder of the baseline models with image reconstruction is based on SAC-AE [16] and has a single-layer fully connected neural network followed by four transpose convolutional layers of $32 \times 32 \times 3$ with ReLU activation function. The stride of the transpose convolutional layers are 1, except for the last layer which has stride 2.

Table 1: Hyperparameters used in our experiments.

| Hyperparameter | Setting |
|---|---|
| Learning rate | 1e−4 |
| Optimizer | Adam |
| $n$-step return | 3 |
| Mini-batch size | 256 |
| Actor update frequency $d$ | 2 |
| Target networks update frequency | 2 |
| Target networks soft-update $\tau$ | 0.01 |
| Target policy smoothing stddev. clip $c$ | 0.3 |
| Hidden dim. | 256 |
| Replay buffer capacity | $10^6$ |
| Discount $\gamma$ | 0.99 |
| Seed frames | 4000 |
| Exploration steps | 2000 |
| Exploration stddev. schedule | $\mathrm{linear}(1.0, 0.1, 1e6)$ |

Table 2: Hyperparameters specific to state observations.

| Hyperparameter | Setting |
|---|---|
| Feature dim. | Same as the state dim. of the task |
| Action repeat | 1 |
| Frame stack | N/A |

Table 3: Hyperparameters specific to pixel observations.

| Hyperparameter | Setting |
|---|---|
| Feature dim. | 50 |
| Action repeat | 2 |
| Frame stack | 3 |

### E.3 Baseline Implementations

All of the baselines are submitted in the supplemental material. We use the official implementations of DBC, SAC-AE, and TD3. DeepMDP does not have a publicly available code, and we use the implementation available in the official DBC code-base. The official DDPG implementation is in TensorFlow, thus we used the implementation available in the official TD3 code-base with additional improvements detailed in Section 7.1. Similarly, the official SAC implementation is in TensorFlow, thus we used the SAC implementation available in the official SAC-AE code-base. As discussed in Section 7.2, we have run two versions of the baselines, with and without image augmentation. The image augmented variants, use the same image augmentation method of DrQ-v2 described in Appendix E.1.