# OpenReview forum: "Continuous MDP Homomorphisms and Homomorphic Policy Gradient"
_NeurIPS.cc/2022/Conference — NeurIPS 2022 Accept_

### Official Review · Reviewer_EAfi · 2022-07-06

**Rating:** 7
**Confidence:** 4
**Soundness:** 4 excellent
**Presentation:** 3 good
**Contribution:** 4 excellent

**Summary:**

The paper extends the theory of MDP homomorphism abstractions to continuous state and action spaces, and includes a theorem that motivates the use of MDP homomorphisms in actor-critic reinforcement learning agents. The paper proposes an agent that learns an abstraction of the ground environment while learning to solve a particular task. The agent is evaluated on continuous control tasks.


**Questions:**

1. The abstraction components (the state and state-action encoders f and g, the abstract critic
\bar{Q}, the abstract transition predictor \tau and the reward predictor \bar{R}) add additional capacity to your model . What is the number of FLOPS (or the number of parameters if it is difficult to measure FLOPS) in these components compared to a basic actor-critic model? Could the model size be controlled for in the experiments to make the comparison against the baseline fairer?
2. The abstract encoders f(s) and g(s, a) could capture similar information, since they both have access to the state. In the extreme, the model could use f(s) for reward prediction only and g(s, a) for everything else, since all functions except for the reward predictor are conditioned on the outputs of both f and g. Do f and g output similar features?
3. Could you clarify lines 305-307 and Figure 6? Please see my comment above.
4. What is the size of the state and the state-action encoding in Figure 5.


**Limitations:**

Limitations are not explicitly discussed, but shoud be similar to previous works such as DBC and DDPG.

**Strengths And Weaknesses:**

# Strengths:

1. The paper extends the theory of MDP homomorphisms to continuous action spaces (in both the ground and the abstract MDP), which I consider to be a significant contribution.
2. The paper aligns with prior literature on using bisimulation metrics in model-free learning by using the lax bisimulation metric, which includes action abstractions. I find this to be well-motivated.
3. The proposed method is evaluated on many continuous control tasks, the high variance in the results of deep reinforcement learning agents is controlled for and the source code for the method is included.

# Weaknesses:

1. There is a possibility that the quantitative benefits we see in the proposed method are due to increased model capacity only. I could not find a discussion of the sizes of the evaluated models.
2. Having an abstraction component should provide additional benefits, such as compactness of representations and robustness to distracting features. These benefits are demonstrated to a limited extent, and some details of the experiments are unclear (please see my detailed comments).

# Detailed comments:

## Writing

– This paper is close to previous works on using bisimulation metrics in model-free reinforcement learning. I am missing a discussion of the similarities between the proposed model and prior works (like DBC [10]) and a discussion of the benefits of adding action abstraction to this framework.

– The literature review is insufficient, many prior works on MDP homomorphisms are missing [1, 2, 3, 4, 5, 6]. Additionally, there are a few more works on bisimulations that could be of interest [7, 8, 9]. This list of papers I provided is non-exhaustive.

– Minor comment: This might come down to personal preference, but I do not like the way the introduction is written. The introduction should tell a story and build interest in the rest of the paper. Instead, the introduction in this paper reads like an extended list of contributions.

– Minor comment: The authors cite one of Ravindran’s original papers on MDP homomorphisms, but not his thesis, which I would consider to be the definitive source [11].

## Experiments

– Rather than an increase in agent performance, which could be attributed to the increase in its capacity, I see the main benefit of state and action abstractions in (a) learning compact representations, (b) ignoring distractions and (c) facilitating better transfer of the learned representation. (a) is shown in the cartpole environment and I find this experiment very compelling. My only complaint is that I could not find what the size of the state-action encoding is (i.e. we could make the state encoding small, but capture all information in the state-action encoding). (b) Is not included in the paper and I find this to be a missed opportunity. For example, DBC [10], which is a baseline in this paper, shows convincing experiments with distracting backgrounds in their Figures 3 and 4. (c) is shown in the appendix, but I am not sure how to interpret the results of the experiment.

– Lines 305 - 307: Perhaps I misunderstand, but the claim that DHPG “mapped all legs to the same abstract latent state” is problematic. Firstly, I do not know if Figure 6b supports this claim. There are four clusters in Figures 6a and 6b, and they look similar. Secondly, it seems unlikely that having a single abstract latent state to track the state of all four legs is desirable – the legs have to move independently to facilitate locomotion. I suppose there is some division of what is represented in the state and the state-action encoder, but that is not explained in the paper.

## References

[1] Vishal Soni and Satinder Singh. 2006. Using Homomorphisms to Transfer Options Across Continuous Reinforcement Learning Domains. In Proceedings of the 21st National Conference on Artificial Intelligence - Volume 1 (AAAI’06). AAAI Press, 494–499.

[2] Alicia P. Wolfe. 2006. Defining Object Types and Options Using MDP Homomorphisms.

[3] Alicia Peregrin Wolfe and Andrew G. Barto. 2006. Decision Tree Methods for Finding
Reusable MDP Homomorphisms. In Proceedings of the 21st National Conference on Artificial Intelligence - Volume 1 (AAAI’06). AAAI Press, 530–535.

[4] Jonathan Sorg and Satinder Singh. 2009. Transfer via Soft Homomorphisms. In Proceedings of The 8th International Conference on Autonomous Agents and Multiagent Systems - Volume 2 (AAMAS ’09). International Foundation for Autonomous Agents and Multiagent Systems, Richland, SC, 741–748.

[5] S. Rajendran and M. Huber. 2009. Learning to generalize and reuse skills using
approximate partial policy homomorphisms. In 2009 IEEE International Conference
on Systems, Man and Cybernetics. 2239–2244.

[6] Ondrej Biza and Robert Platt. 2019. Online abstraction with MDP homomorphisms for deep learning. In International Conference on Autonomous Agents and MultiAgent Systems.

[7] Abel, D.; Hershkowitz, D. E.; and Littman, M. L. 2016. Near Optimal Behavior via Approximate State Abstraction. In Proceedings of the International Conference on Machine Learning, 2915–2923.

[8] Abel, D.; Arumugam, D.; Asadi, K.; Jinnai, Y.; Littman, M. L.; and Wong, L. L. S. 2019. State Abstraction as Compression in Apprenticeship Learning. In AAAI.

[9] Ondrej Biza, Robert Platt, Jan-Willem van de Meent, and Lawson LS Wong. 2020. Learning Discrete State Abstractions with Deep Variational Inference. In Proceedings of the 3rd Symposium on Advances in Approximate Bayesian Inference.

[10] https://arxiv.org/abs/2006.10742

[11] https://all.cs.umass.edu/pubs/2004/ravindran_thesis04.pdf

---

> ### Author Response · Authors · 2022-08-02
> **Response to Reviewer EAfi**
>
> Thank you for your review.
>
> ## Weaknesses
> 1. [Reviewer asks about the increased network capacity] Thank you for pointing this out. This is a very valid concern and we have tried to address it by giving a detailed architecture in Appendix E.2 and submitting our code. Compared to DBC, our method requires a parameterized MDP homomorphism and an abstract critic (both are MLPs with 2 hidden layers of 256). Therefore, we do not believe that the performance boost is accredited to the increased capacity of the models. To confirm this, we added controlled experiments to the revised paper in **Section 7**, and added **Appendix D.7**. Please see the revised paper and our response to Q1 below.
> 2. [Reviewer asks for more empirical results] While we completely agree with your points, we also believe that abstraction allows for more data-efficient learning. A key algorithmic benefit of the HPG theorem is data efficiency, as we can use an additional policy estimator using samples mapped via the MDP homomorphism map. Nevertheless, we do agree that rigorously evaluating the quality of the learned MDP homomorphism is important. Therefore, in addition to our empirical evidence on the compactness of the representations (Fig 5) and transferability of the representations  (Appendix D.3), we have evaluated the quality of the learned MDP homomorphism maps in Appendix D.4, by analyzing the value equivalence between the abstract and actual critics. Please see **point #2** in our general response for DMC with distractions.
> ## Detailed Comments
> - Comparison with DBC: Please see **point #1** in our general response. We added this comparison to Section 1.
> - References: Thank you for pointing out the missing references. We added them to the revised version.
> - Experiments: Please see point #2 of the weaknesses above for data-efficiency, Q4 below for the size of the state-action encoder, and **point #2** in our general response for DMC with distractions. Regarding the transferability of the representations, our goal is to show that a learned MDP homomorphism is transferable to a new reward setting within the same domain. This is mainly to address the common concern regarding transferability of representations that rely on the reward function during training. We added a clarification on this in the revised paper.
>
> ## Questions
> 1. [Reviewer asks for empirical results with controlled network capacity] We report the number of parameters for DHPG, DBC, and DrQv2 for the cartpole env with a 50 dimensional latent space:
>    * DHPG: image encoder (1990518) + actor (79105) + critic (79361) + dynamics model (117348) + reward model (79105) + abstract critic (91905) + f (91698) + g (91954) = 2,620,994
>    * DBC: image encoder (1990518) + actor (79362) + critic (158722) + dynamics model (104804) + reward model (79105) = 2,412,511
>    * DrQ-v2: image encoder (1990518) + actor (79105) + critic (158722) = 2,228,602
>
>    The majority of the parameters are due to the image encoder which is a common component between all models. Note that DHPG uses the vanilla DDPG critic while DBC and DrQ use the TD3 critic. Notably, the increase in parameters of DHPG is due to the parameterization of the MDP homomorphism map rather than actor/critic networks which is itself a sign for the particular effectiveness of the HPG theorem.
>
>    To account for the parameter increase, we ran variations of DBC and DrQ with a larger critic (512 hidden dim compared to the initial 256). Consequently, the new total number of parameters for DBC and DrQ are respectively 2,833,375 and 2,649,466 (both larger than DHPG). Please see **Section 7** and **Appendix D.7** in the revised paper; the results show that DHPG outperforms or matches the performance of the higher-capacity variants of the baselines.
>
> 2. We may have misunderstood your question, but since the output of f is an abstract state and g is an abstract action, they are of different natures. Particularly, since they are both needed independently for the policy lifting process and performing the HPG update, we need to have two separate networks to parameterize them.
> 3. [Reviewer asks for clarification on Fig 6] Please see **point #4** in our general comments. We edited the caption in the revised paper.
> 4. [Reviewer asks for details of Fig 5] The cartpole environment has 4 states and 1 action. Therefore, in Fig 5, all algorithms are limited to a 4D latent space; that is the output of the image encoder is a 4D vector. DHPG maps this latent space to a 4D abstract state space using f, and maps the 5D state-action (4 for latent state and 1 for action) onto a 1D abstract action using g.
>
> ## Limitations
> Yes, the limitation of our deep RL algorithm is similar to DDPG and DBC. But in particular, we have discussed the limitations of our theoretical results (deterministic policies, bijection of g) in Section 4 and the limitations of our empirical results (transferability of representations) in Section 7.

---

> > ### Comment · Reviewer_EAfi · 2022-08-05
> > **Response to Rebuttal**
> >
> > Thank you for the additional work you have done for the rebuttal!
> >
> > When the number of parameters in DHPG and DrQ-v2 is roughly equal, it seems their performance is also about equal (I believe the figure that shows this was moved into the supplementary material). It is somewhat surprising that increasing the number of parameters in DrQ-v2 by about 20% leads to major gains in performance.
> >
> > Since DHPG uses the same hyper-parameters as DrQ-v2 (per the appendix), my interpretation of the results is that abstraction is not needed in the original DM Control environments. Therefore, it is important to show results in the setting with distractors. I sympathize that these experiments take a long time to run.

---

> > > ### Author Response · Authors · 2022-08-06
> > > **Detailed Comparison with Higher-Capacity Baselines and Restatement of Our Theoretical Contributions**
> > >
> > > Thank you for your reply.
> > >
> > > As acknowledged in our general response, we agree with the addition of experiments with DMC with distractions and they will be ready for the camera-ready version. However, we politely disagree with your statement regarding the comparison with DrQ-v2 for the following reasons:
> > > * In Figure 12 in the Appendix, DHPG clearly outperforms the higher-capacity variant of DrQ-v2 in environments with clear symmetries, such as cartpole (particularly the sparse swingup task), pendulum, cup catch, and finger spin. This shows that DHPG can leverage the environmental symmetries for representation learning and better data efficiency.
> > > * DHPG outperforms the higher-capacity variant of DrQ-v2 at the 500k step mark, shown in Figure 13a and 14a in the Appendix, whereas DrQ-v2 slightly outperforms DHPG at the 1m step mark. This shows that overall DHPG has a better data-efficiency.
> > >
> > > Additionally, as shown in Figure 7 of the main body and Figures 13-14 of the Appendix, we would like to point out that DHPG (with 2.6m parameters) significantly outperforms the higher-capacity variant of DBC (with 2.8m parameters) which yet again shows the benefit of our algorithm that is theoretically motivated by the novel homomorphic policy gradient (HPG) theorem.
> > >
> > > Once again, we would like to remark the significance of our theoretical contributions, namely extending the theory of MDP homomorphisms to continous states and actions and the derivation of the homomorphic policy gradient theorem. We believe that our work will
> > > open-up future possibilities for MDP homomorphisms in challenging continuous control problems.
> > >
> > > Finally, we would like to remark that in addition to adding the requested experiments on the higher-capacity variants, we believe we have addressed every comment and request made by the reviewers. We hope that this builds some good faith, that if the paper were to be accepted, the experiments with distractions (which are more than x5 slower) will be added to the camera-ready version.

---

> > > > ### Comment · Reviewer_EAfi · 2022-08-08
> > > > **Response**
> > > >
> > > > Thanks for the clarification! I agree that this paper has a significant theoretical contribution and favorable experimental results, so I am definitely in favor of acceptance. On the topic of DHPG vs DrQ-v2, it is also very possible that DHPG could be tuned to get even better results, whereas DrQ-v2 is already highly tuned.

---

> > > > > ### Author Response · Authors · 2022-08-08
> > > > > **Thank you**
> > > > >
> > > > > Thank you for acknowledging our theoretical contributions and re-adjusting your rating! We agree on the comparison with DrQ-v2, as we have used all of the hyperparameters of DrQ-v2 without any further tuning to DHPG.

---

### Official Review · Reviewer_c7GW · 2022-07-09

**Rating:** 7
**Confidence:** 3
**Soundness:** 4 excellent
**Presentation:** 3 good
**Contribution:** 3 good

**Summary:**

This paper proposes MDP homomorphisms and homomorphic policy gradient for continuous control tasks. The contributions are threefold: 1. introducing the continuous MDP homomorphisms, 2. deriving the homomorphic policy gradient theorem that ties the induced abstract MDP into the policy gradient, 3. developing the deep homomorphic policy gradient algorithm which simutaneously learn the policy and homomorphism.

**Questions:**

1. In Line 165, the authors mentioned "we assume having access to an MDP homomorphism map ... . The problem of learning such mapping from samples is addressed in Section 6". Would the derived policy gradient and the policy optimisation algorithm depend on an accurate MDP homomorphism?
2. In addition to the first question, what happens to Theorem 4 and 5 if the MDP homomorphism is not an exact equality (as in Equations (6) and (7) where there is an epsilon difference) [1]?

[1] Abel, David, David Hershkowitz, and Michael Littman. "Near optimal behavior via approximate state abstraction." International Conference on Machine Learning. PMLR, 2016.


**Limitations:**

The authors clearly discussed the limitation of assuming deterministic policies in Section 4.2.

**Strengths And Weaknesses:**

1. The introduction of MDP homomorphisms to continuous control tasks is interesting and significant.
2. The paper is overall well written, with clear motivations and theoretical explainations.
3. The authors agree to make their code publicly available and has provided the source code in the appendix.

---

> ### Author Response · Authors · 2022-08-02
> **Response to Reviewer c7GW**
>
> Thank you for your review.
>
> ## Questions
> 1. [Reviewer asks for the validity of our theoretical results under an approximate MDP homomorphism map] Thank you for pointing this out; this is an excellent point. Yes, our derivation for the  homomorphic policy gradient theorem assumes an accurate MDP homomorphism. In fact, extending the theoretical results to approximate MDP homomorphisms is a very interesting next step for our future work that requires intricate analysis .
>
>    Regardless of our theoretical results, our empirical results suggest that an approximate MDP homomorphism map is still very valuable and our algorithm can leverage such an approximation to update the actual policy. In practice, since we are representing the MDP homomorphism $\(f, g_s)$ by neural networks and we are training it with samples, the MDP homomorphism is inevitably an approximation even for environments with perfect symmetries.
>
>    Finally, to empirically analyze the quality of the learned approximate MDP homomorphism maps, we have plotted the value equivalence between the abstract and actual critics in Figure 8 in Appendix D.4. As the results suggest, the value equivalence property (which is needed to derive our homomorphic policy gradient theorem) is holding in practice with a reasonable amount of error. As the training progresses and the MDP homomorphism map is trained, this error decreases.
>
> 2. [Reviewer asks for the validity of our theoretical results under an approximate MDP homomorphism map] Thank you for your question; similarly to Q1, this is a great point. Theorems 4 and 5 hold for exact MDP homomorphisms and extending their results to approximate MDP homomorphisms requires a substantial amount of non-trivial work, something that is very interesting to us for our future directions.
>
>    Notably, the original lax bisimulation paper [1] derives an upper bound for the optimal value equivalence property (see for example Theorem 14 in [1]). Our first step is to extend this result to the value equivalence property (Theorem 1 in our paper) and proceed to the derivation of the homomorphic policy gradient theorem.
>
> [1] Taylor, Jonathan, Doina Precup, and Prakash Panagaden. "Bounding performance loss in approximate MDP homomorphisms." Advances in Neural Information Processing Systems 21 (2008).

---

> > ### Comment · Reviewer_c7GW · 2022-08-08
> > **Response to Rebuttal**
> >
> > Thanks for the insights on the questions regarding approximate MDP homomorphism map. I acknowledge the authors' comments that extending the theoretical results to approximate cases would be a non-trivial extension which can be a future step. It is nice to see the empirical results in the present paper show the potential of the proposed formulation on approximate homomorphism cases.

---

> > > ### Author Response · Authors · 2022-08-08
> > > **Thank you**
> > >
> > > Thank you for your reply and acknowledging that extending the theoretical results to approximate MDP homomorphism can be a future step.

---

### Official Review · Reviewer_8n1B · 2022-07-10

**Rating:** 7
**Confidence:** 3
**Soundness:** 3 good
**Presentation:** 3 good
**Contribution:** 3 good

**Summary:**

This paper contributes a) a continuous version of the MDP homomorphism and b) a homomorphic policy gradient. An approach based on lax bisimulation metrics learns a policy and a continuous MDP homomorphism simultaneously. The paper evaluates on DM control tasks from pixel observations and shows improvements compared to reasonable baselines (DrQ-v2, DBC, DeepMDP, etc).

**Questions:**

- Could you elaborate on what the effect is of requiring $g_s$ to be a bijection?
- In Figure 3a, the learned policy looks approximately, but not exactly symmetric. The original problem does have exact symmetry, and in Fig 3b the action representations look like they may have recovered the symmetry. Why is the policy less symmetric than the representations are? (This Figure is hard to read. It would be improved by clarifying which of the plotted values represent the chosen actions)
- The paper claims that legs are mapped onto each other in Figure 6a/b. Could you elaborate on how one can read this in the Figure and how you know that legs are represented in the abstract/latent state space structure?
- Suggested references:
    - Park, Jung Yeon, et al. “Learning Symmetric Embeddings for Equivariant World Models.” ICML 2020
    - Narayanamurthy, Shravan Matthur, and Balaraman Ravindran. “On the hardness of finding symmetries in Markov decision processes.” ICML 2008.
- Appendix C.1 says that $R$ and $\bar{R}$ are both bounded. Where does this assumption come from?
- Page 2, line 50-51 claims that the proposed approach leads to more robust solutions. The approach seems similar to an MDP homomorphic version of DBC. Why does the approach outperform DBC?


**Limitations:**

- The checklist says empirical limitations are discussed in Section 7, but I could not find this. Could you point me towards the paragraphs?
- Section 4 discusses part of the theoretical assumptions/limitations.
- Code was included with submission.


**Strengths And Weaknesses:**

# Strengths
- The paper tackles an important problem: learning representations for reinforcement learning problems, especially those that recover structure and/or symmetry in the input MDP.
- The approach, of mapping into an abstract space and learning a policy + abstraction simultaneously is reasonable and has precedent.
- According to the paper, the approach is able to learn in smaller latent dimensions and recover symmetric representations, which is a nice result. Discovering symmetries in MDPs is a largely unsolved problem, and the paper takes a step in that direction.
- It's useful to have an extension to continuous MDP homomorphisms and to have the homomorphic policy gradient theorem. The paper could be of interest to the field of state representation learning and to the field of symmetries in RL.

# Weaknesses
- Usually in MDP Homomorphisms, $g_s$ is a surjection. In the paper, it is required to be a bijection. This comes with the disadvantage that equivalent actions belonging to a single state can no longer be aggregated/mapped onto the same abstract action. For symmetries this should not be an issue, but making $g_s$ bijective limits the levels of aggregation in action space.
- The work is evaluated on DMControl tasks. One of the baselines is DBC. In the original DBC paper and follow-up work (e.g. [42]), there are experiments with background distraction videos. Including such experiments in this work, to show robustness to distractors that are irrelevant to the task, would make the paper stronger.
- The potential of this approach to find symmetries in MDPs is part of its appeal. The paper does allude to this by including plots that seem to show some degree of symmetry recovery. However, the direction is not explored in-depth in the paper.
- Missing references:
    - Mahajan, Anuj, and Theja Tulabandhula. “Symmetry learning for function approximation in reinforcement learning.” arXiv preprint arXiv:1706.02999 (2017).
    - Biza, Ondrej, and Robert Platt. “Online abstraction with MDP homomorphisms for deep learning” AAMAS 2019

---

> ### Author Response · Authors · 2022-08-02
> **Response to Reviewer 8n1B**
>
> Thank you for your review.
>
> ## Weaknesses
> 1. **Bijection assumption of** $g_s$: Thank you for pointing this out; this is an important point. Yes, the bijection of $g_s$ prevents us from aggregating actions, as you mentioned. In general, the policy lifting process for deterministic policies is defined as selecting one representative for the preimage $g_s^{-1}(\bar{\pi}(f(s)))$, however, since there may be multiple choices that satisfy this condition (due to the surjection of $g_s$), we have assumed $g_s$ to be a bijection. Therefore the deterministic lifted policy can be uniquely obtained as $\pi^\uparrow (s) = g_s^{-1}(\bar{\pi}(f(s)))$. We require this uniqueness for our subsequent homomorphic policy gradient theorem. Specifically, we use the inverse function theorem to prove the equivalence of deterministic policy gradients in Theorem 4.
>
>     Notably, the bijection assumption of $g_s$ is only required for our theoretical results to hold. In practice, our DHPG algorithm does not enforce the action encoder $g_s$ to be an invertible neural network. Therefore, action aggregation can happen in practice. Finally, our preliminary theoretical results for the homomorphic policy gradients for stochastic policies (which is an ongoing work) indicates that $g_s$ is not required to be a bijection for stochastic policies, which may be of interest in certain MDPs. We did not include these results in the present paper as it is a major change in all the proofs and is best developed separately. Please see the answer for Q1 below for more details.
>
> 2. **DM control with distractions:** Thank you for your suggestion, please see **point #2** in our general comments.
>
> 3. **Symmetry recovery in MDPs:** We agree that this is a very interesting aspect to further showcase the abilities of our method. However, analyzing and visualizing such symmetries, particularly for high-dimensional continuous MDPs is quite challenging. Therefore, we chose two domains (pendulum and quadruped) as informative examples, but we are looking forward to hearing your suggestions for further analysis of such symmetries in continuous MDPs.
>
> ## Questions
> 1. [Reviewer asks for elaboration on the bijection assumption of $g_s$] As you pointed out, the bijection assumption of $g_s$ prevents us from aggregating actions in theory. However, in practice since this bijection is not enforced, the action aggregation may happen. Another implication of this assumption is that the actual action space and the abstract action space need to have the same dimensionality, which in turn prevents us from removing redundant dimensionalities from the action space. This arises in MDPs with continuous symmetries; a simple example is an extension of the 2D mountain car to the 3D space in which case the third dimension has a translational symmetry. Thus, actions along that dimension can be thought of to be redundant.
>
> 2. [Reviewer asks for clarification of Fig 3] Thank you for raising this; we updated the caption of Fig 3 for clarification. Fig 3a visualizes the actual optimal policy by plotting the contours of the selected actions over the state space. The data of Fig 3a is generated by rolling out the actual optimal policy for some number of episodes. Fig 3b visualizes the abstract optimal policy by plotting the contours of the abstract actions over the state space. The data of Fig 3b is obtained by passing state-action pairs of Fig 3a through the action encoder $g_s$. Notably, the actual optimal policy is not symmetric, and it is in fact not expected to be symmetric (because the optimal actions for two equivalent states have opposite signs). However, the abstract optimal policy is symmetric with respect to the actual state-space, as observed in Fig 3b.
> 3. [Reviewer asks for clarification of Fig 6] Please see **point #4** in our general comments. We edited the caption and the text around Fig 6 in the revised version.
> 4. [Reviewer suggests references] Thank you for your suggested references. We have added them to the revised version.
> 5. [Reviewer asks for the assumption of boundedness of reward] The boundedness of the reward function is a standard assumption in the literature as it makes the discounted return to be a finite amount. We have listed this as our assumptions in Appendix B. Some examples of this assumption are:
>    * Section 3.3 of Sutton and Barto, 2nd Edition.
>    * Regularity conditions in Appendix A of the original DPG paper (Silver, David, et al. "Deterministic policy gradient algorithms." 2014.)
> 6. [Reviewer asks for detailed comparison with DBC] Please see **point #1** in our general response. We added this comparison to Section 1.
>
> ## Limitations
> [Reviewer acknowledges our statement of theoretical limitations but asks for empirical limitations] We discuss the possible limitations of transferring representations to new rewards within the same domain in L308-313 of the initial submission (or L314-L320 of the revised paper).

---

### Author Response · Authors · 2022-08-02
**General Response to All Reviewers**

We thank all of the reviewers for their constructive and helpful feedback. We appreciate that they have found our paper interesting and have acknowledged the significance of our contributions and the soundness of our theoretical and empirical results.

The modified paper, posted here, addresses the reviewers’ insightful questions by improving our comparison to DBC, adds additional experiments and clarifies writing.

## Response to common comments and questions
1. **Comparison with DBC**: While our practical deep RL algorithm may appear similar to DBC on the surface, motivated by our theoretical results, we have replaced the bisimulation metric of DBC with the lax bisimulation metric and have added an action encoder for abstracting actions. This is quite a significant change as it allows us to exploit symmetries in the MDP dynamics. We draw two detailed distinctions from DBC:
    * **HPG-joint updates:** One of our main contributions is the homomorphic policy gradient (HPG) theorem, which allows for direct optimization of the actual policy using the abstract MDP. DBC lacks this theoretical grounding. A similar policy gradient was not derived for the bisimulation relation. Our theoretically motivated HPG solution updates representations and policy jointly, while DBC separated these steps. We believe that this is the key to our performance boost over DBC.
    * **Class of symmetries:** Bisimulation does not allow abstracting actions, while this is a key component of homomorphisms, giving them greater modeling flexibility. An elementary example is a pendulum, where right and left halves are typically not bisimilar, because the opposite action is needed. By relabeling both states and actions, they are homomorphic. We believe that this is a key strength of our approach, allowing it to capture many more symmetries and equivariances for fast and robust learning, as supported by our results.

2. **DM control with distractions:** We thank the reviewers for their suggestion. We have started these experiments and they will be added to the final version. Notably, these experiments take a considerably longer time (more than x5 slower) due to the additional operations on the images and our commitment to providing a thorough and rigorous empirical study on 10 seeds for all algorithms on all environments has unfortunately prevented us from running DM control with distractions for the initial submission.

3. **Missing references:** We thank the reviewers for pointing out the missing references. All of them, as well as additional references, have been added to the revised version of the paper.

4. **Clarification on Figure 6:** Thank you for raising this; we agree that Fig 6 may be hard to interpret. First, please note that analyzing and visualizing symmetries in high-dimensional continuous MDPs is quite challenging and we are looking forward to hearing your suggestions on this. This is in contrast to Fig 3 which is on a much simpler MDP (pendulum) and shows clear signs of symmetries within the abstract MDP. Fig 6a shows four distinct branches (in light blue) in the latent states; we believe that these branches are corresponding to different configurations of the four legs. While Fig 6b shows that these four branches (light blue) are closely aligned together in the abstract MDP and have created two distinct branches. We hypothesize that this is due to the abstraction resulting from the MDP homomorphism map. For example, moving the left forward leg is equivalent to moving the right backward leg, and moving the right forward leg is equivalent to moving the left backward leg. We agree with **EAfi**, and have reworded the explanation in the revised paper.

## Summary of the changes in the revised paper
All changes are in the red color. Please note that the final version, upon acceptance, is allowed to have 10 pages.
1. Added new experiments to Section 7 and added Appendix D.7 as requested by **EAfi**. These experiments aim to address concerns regarding the increased network capacity of DHPG compared to the baselines.
2. Added all of the missing references suggested by the reviewers.
3. Added a more clear comparison with DBC and other related works in Section 1 (L47-L59), as requested by **8n1B** and **EAfi**.
4. Edited the caption of Figure 3, as requested by **8n1B**.
5. Edited the caption of Figure 6, as requested by **8n1B** and **EAfi**.
6. Added a clarification on transfer learning experiments in Section 7, as requested by **EAfi**.

---

> ### Author Response · Authors · 2022-08-05
> **Update on the Revision**
>
> We realized that, per the NeurIPS guidelines, the revised paper during the rebuttal should follow the 9-page limit. Therefore, we have updated the revised paper with a subset of the changes listed above and have included the 10-page revised paper in the supplementary zip file. Please refer to the supplementary material for the revised version that includes additional experimental results and clarifications.
>
> We apologize for any inconvenience this may have caused.

---

### Meta-Review · Area_Chair_VyyP · 2022-08-26

**Recommendation:** Accept
**Confidence:** Certain

**Metareview:**

The paper a new algorithm for representation learning in RL based on MDP homomorphism. The paper presents a nice theory, the algorithm is well motivated and the experiments are convincing. The concerns of the reviewers such as comparisons with DBC or DM control suite with distractions have been addressed adequately by the authors. All reviewers evaluated the paper very positively and I join their decision.

**Award:**

No

---

### Decision · Program_Chairs · 2022-09-14

Accept